# Rebalancing Contrastive Alignment with Bottlenecked Semantic Increments in Text-Video Retrieval

**Jian Xiao**[1], **Zijie Song**[2], **Jialong Hu**[1], **Hao Cheng**[1], **Zhenzhen Hu**[1*], **Jia Li**[1*], **Richang Hong**[1]

[1]School of Computer Science and Information Engineering, Hefei University of Technology, Hefei, China
[2]School of Big Data and Statistics, Anhui University, Hefei, China

{j.xiao_hfut, chenghao}@mail.hfut.edu.cn, zjsong@ahu.edu.cn
zdszds534@gmail.com, {zzhu,lijia}@hfut.edu.cn, hongrc.hfut@gmail.com

## Abstract

Recent progress in text–video retrieval has been largely driven by contrastive learning. However, existing methods often overlook the effect of the modality gap, which causes anchor representations to undergo in-place optimization (i.e., optimization tension) that limits their alignment capacity. Moreover, noisy hard negatives further distort the semantics of anchors. To address these issues, we propose GARE, a Gap-Aware Retrieval framework that introduces a learnable, pair-specific increment $\Delta_{ij}$ between text $t_i$ and video $v_j$, redistributing gradients to relieve optimization tension and absorb noise. We derive $\Delta_{ij}$ via a multivariate first-order Taylor expansion of the InfoNCE loss under a trust-region constraint, showing that it guides updates along locally consistent descent directions. A lightweight neural module conditioned on the semantic gap couples increments across batches for structure-aware correction. Furthermore, we regularize $\Delta$ through a variational information bottleneck with relaxed compression, enhancing stability and semantic consistency. Experiments on four benchmarks demonstrate that GARE consistently improves alignment accuracy and robustness, validating the effectiveness of gap-aware tension mitigation. Code is available at https://github.com/musicman217/GARE-text-video-retrieval.

## 1 Introduction

Text-video retrieval (TVR) [55] is a fundamental task in video understanding, aiming to retrieve relevant videos given a text query [34, 33, 45, 17]. With the proliferation of short video platforms, this task has attracted growing research interest. In recent years, vision-language pretraining models such as CLIP [38] have shown great success in cross-modal representation alignment, demonstrating strong performance on various retrieval benchmarks. These models typically learn a shared embedding space by aligning visual and textual modalities through large-scale contrastive learning [50, 20, 10, 11, 19, 46, 22, 31], and have thus become a popular backbone in TVR systems.

Despite the empirical success of contrastive learning in text-video retrieval, two critical problems persist. First, the most challenge is optimization tension, arising from the modality gap: text and video embeddings typically occupy disjoint regions of the representation space [31, 54], with markedly different semantic structures and high distributional divergence (e.g., large KL divergence [28] between modality-wise feature distributions). As shown in Figure 1a, this separation creates a structural conflict for a text anchor $t_i$: the gradient from its positive video $v_i$ attracts $t_i$ toward the video manifold, while gradients from all negatives $v_j$ repel it in the opposite direction, yielding nearly collinear but reversed forces. The second is the prevalence of false negatives: semantically similar

---

*Corresponding author.

39th Conference on Neural Information Processing Systems (NeurIPS 2025).

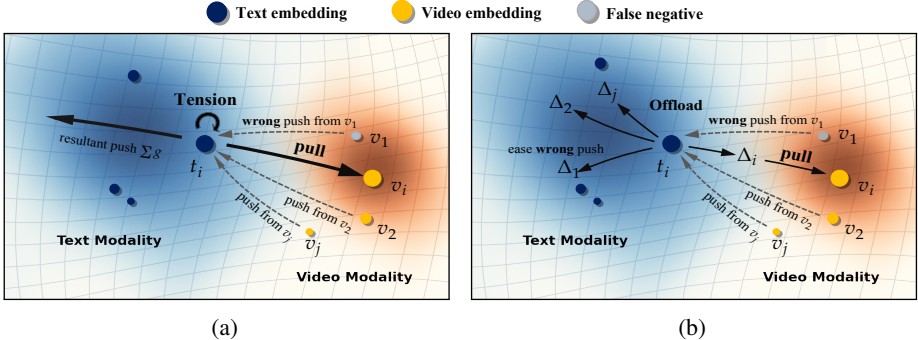

Figure 1: **Tension and false-negative challenge vs. our offloading strategy.** (a) Owing to the modality gap [31], gradients from negative samples overlap with the positive direction, creating optimization tension around the anchor $t_i$ and limiting its update freedom. (b) GARE offloads part of this optimization pressure from $t_i$ to learnable increments $\Delta_{ij}$, relaxing the gradient field and absorbing false-negative noise. Each $\Delta_{ij}$ encodes a semantically meaningful correction of the text–video gap.

yet unlabeled pairs are incorrectly treated as hard negatives [12, 9], introducing noisy gradients and aggravating misalignment. These two issues jointly limit the ceiling of semantic alignment, leading models to converge to suboptimal solutions.

We further analyze the optimization tension through aggregated gradient statistics across batches (Figure 2). In dimensions exhibiting significant gradient behavior, both positive and negative components reach values on the order of 40–60 (bottom of Figure 2), yet their sum—the actual gradient applied to $t_i$—remains close to 2–3 (top of Figure 2). This indicates that positive and negative signals not only oppose each other in direction but also nearly cancel in magnitude. As a result, text anchors undergo an in-place optimization behavior: their representations remain close to their initial positions throughout training, thereby limiting the attainable alignment performance of contrastive learning.

To address both issues, we introduce a pair-specific increment $\Delta_{ij}$ that acts as a learnable adjustment between $t_i$ and $v_j$. As shown in Figure 1b, unlike the anchor embedding $t_i$, which aggregates gradients from all video pairs $(i, k)$, the increment $\Delta_{ij}$ only absorbs gradients transmitted from its corresponding pair $(i, j)$. This ensures that each $\Delta_{ij}$ captures a localized component of the optimization signal, while $t_i$ retains global supervision. As a result, gradients acting on $t_i$ are partially

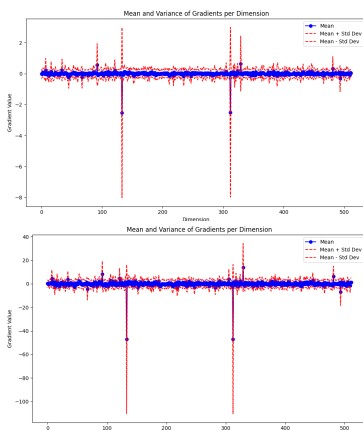

Figure 2: Mean and variance of summed gradient (top) and negative gradients (bottom) across 512 dimensions, showing collinear but opposite forces that largely cancel out.

redistributed to $\Delta_{ij}$, effectively diluting the optimization tension and preventing anchors from being trapped in conflicting descent directions. Beyond relieving tension, this design also buffers the local gradient noise introduced by false negatives, since noisy repulsion from mislabeled pairs is absorbed at the pair level rather than directly perturbing anchor representations.

To guide $\Delta_{ij}$ toward constructive updates, we derive its local rule from a multivariate first-order Taylor expansion of the contrastive loss under an $\ell_2$ trust-region constraint. This linearization defines a descent space in which, once a constraint radius is specified, the steepest-descent direction at the coupled state becomes unique and preserves the local relative-ranking structure of InfoNCE [37]. We implement this process as a pair-specific, amortized update using a lightweight module $\psi$ conditioned on the semantic gap $(v_j - t_i)$. The module is optimized through backpropagation across batches, while a norm-based prior regularizes the magnitude of $\Delta$, serving as an implicit trust-region radius regularization. To ensure semantically stable and generalizable updates, we formulate the learning of $\Delta_{ij}$ as a deterministic variational information bottleneck: the module $\psi$ outputs $\Delta_{ij}$, and a Jensen-relaxed KL bottleneck term balances informativeness and compression.

On the MSR-VTT 1k-A validation set, we observe that the overall cosine similarity of positive pairs decreases relative to the baseline. This indicates that releasing optimization tension via $\Delta$ encourages samples to spread with larger angular separations, thereby improving uniformity on the unit hypersphere [46]. In addition, we find that the Euclidean distance between paired embeddings increases after applying $\Delta_{ij}$, suggesting that pair-level adjustments help alleviate tension and provide finer control over semantic positioning.

Our contributions are threefold: 1) We analyze the gradient structure of InfoNCE and reveal its inherent multi-variable coupling by introducing pairwise increments $\Delta_{ij}$. A multivariate first-order Taylor expansion within a trust region yields a update rule for each $\Delta_{ij}$ consistent with the InfoNCE descent direction. 2) We propose a Gap-Aware Retrieval (GARE) framework, where a learnable network predicts pair-specific increments $\Delta_{ij}$ and integrates them into the forward pass to offload optimization tension while mitigating noise from false negatives. We also introduce a relaxed variational information bottleneck (VIB) objective that regularizes $\Delta_{ij}$, balancing informativeness and compression. 3) Experiments on four text–video retrieval benchmarks, i.e., MSR-VTT [52], DiDeMo [2], ActivityNet Captions [27], and MSVD [8], showing consistent improvements, and further analyses confirm that the learned increments are semantically meaningful and geometrically structured.

## 2  Approach

### 2.1  Preliminaries

**Task Definition.**    Given a dataset of $N$ paired video-text examples $\{(v_i, t_i)\}_{i=1}^{N}$, the goal of text-video retrieval is to learn a pair of encoders: a visual encoder $\phi_v(\cdot)$ and a text encoder $\phi_t(\cdot)$, that map inputs into a shared embedding space. The similarity between any pair $(v_j, t_i)$ is computed using a scoring function, typically the cosine similarity:

$$s_{ij} = \frac{\phi_t(t_i)^{\top} \phi_v(v_j)}{\|\phi_t(t_i)\| \cdot \|\phi_v(v_j)\|}, \tag{1}$$

where $\phi_t(t_i) \in \mathbb{R}^D$ and $D$ is dimension. During training, contrastive learning is applied to increase the similarity of matched pairs while decreasing that of mismatched ones. At inference, retrieval is performed by ranking all candidate texts (or videos) for a given query video (or text) based on similarity scores. For brevity, we henceforth refer to $\phi_t(t)$ as $t$, $\phi_v(v)$ as $v$.

**Optimization Tension and First-order Increment Modeling.**    As discussed in the introduction, contrastive optimization in video-text retrieval suffers from gradient tension caused by modality gap and noise from false negatives. These factors hinder stable optimization of the anchor representation $t_i$, which must simultaneously align with the positive $v_i$ and repel all negatives $v_{j \neq i}$. Formally, the per-anchor InfoNCE loss is given by:

$$\mathcal{L}_i = -\log \frac{e^{\cos(t_i, v_i)/\tau}}{\sum_{j=1}^{B} e^{\cos(t_i, v_j)/\tau}}, \tag{2}$$

and its gradient with respect to $t_i$ is:

$$\nabla_{t_i} \mathcal{L}_i = \sum_{j=1}^{B} \underbrace{\frac{(p_{ij} - y_{ij})}{\tau}}_{\text{weight } \frac{\partial \mathcal{L}_i}{\partial s_{ij}} \in \mathbb{R}} \cdot \underbrace{\left[ \frac{v_j}{\|t_i\|\|v_j\|} - \cos(t_i, v_j) \cdot \frac{t_i}{\|t_i\|^2} \right]}_{\text{gradient basis } \frac{\partial s_{ij}}{\partial t_i} \in \mathbb{R}^D}, \tag{3}$$

where $p_{ij} = \mathrm{softmax}(\cos(t_i, v_j)/\tau)$, $y_{ij} = \mathbb{1}_{[j=i]}$, $\tau$ is temperature parameter and $B$ is batch size. Here, the weight term $\frac{\partial \mathcal{L}_i}{\partial s_{ij}} \in \mathbb{R}$ controls the relative strength of each pair, while the gradient basis $\frac{\partial s_{ij}}{\partial t_i} \in \mathbb{R}^D$ specifies the update direction. Under a strong modality gap, the positive and negative gradient bases are nearly collinear but reversed, and the positive weight $|p_{ii} - 1|$ is of comparable magnitude to the aggregate negative weight $\sum_{j \neq i} p_{ij}$ [44]. Together, these conditions cause the gradients to largely cancel along nearly the same axis, leaving only a small residual update for $t_i$.

To alleviate both challenges, we introduce a pair-specific increment $\Delta_{ij}$ that locally adjusts the relative positioning of each text-video pair. This increment serves two purposes: it offloads optimization

tension by redistributing part of the gradients originally acting on $t_i$ to the pair level, and buffers false-negative noise before it propagates to the anchor embedding. For clarity, we first consider applying $\Delta_{ij}$ to the text side, yielding an adjusted representation $t_{\Delta_{ij}} = t_i + \Delta_{ij}$. The same mechanism can also be applied to the video side (e.g., $v_{\Delta_{ij}} = v_j + \Delta_{ij}$), depending on dataset characteristics. This flexibility raises a central question: how should each $\Delta_{ij}$ be optimized to effectively reduce the contrastive loss?

A naive approach would linearize the loss around the original anchor $t_i$ (i.e., $\Delta_{i*} = 0$), treating each $\Delta_{ij}$ as an independent perturbation. However, under this formulation, the gradient $\nabla_{\Delta_{ij}}\mathcal{L}_i$ depends only on the static similarity $\cos(t_i, v_j)$, failing to account for the influence of other $\Delta_{ik}$ in the same batch. This leads to decoupled gradients that ignore the softmax coupling intrinsic to InfoNCE. More importantly, expanding the loss only at $t_i$ results in a univariate approximation, which does not reflect how different $\Delta_{ij}$ collectively reshape the similarity ranking across all $v_j$. Since our goal is to adjust the relative structure of the entire pair set for each anchor, we require a multivariate formulation where all $\Delta_{ij}$ are coupled and optimized under a shared comparison scale.

To this end, we treat all $\Delta_{ij}$ as jointly optimized variables, and reinterpret the per-anchor contrastive loss $\mathcal{L}_i$ as multivariate function over the full set $\{\Delta_{ij}\}_{j=1}^B$:

$$\mathcal{L}_i(\Delta_{i1}, \ldots, \Delta_{iB}) = -\log \frac{e^{s_{ii}/\tau}}{\sum_{j=1}^B e^{s_{ij}/\tau}}, \quad s_{ij} = \cos(t_i + \Delta_{ij}, v_j). \tag{4}$$

Unlike standard per-sample optimization, the softmax structure couples all $\Delta_{ij}$, meaning the gradient of any single increment depends on the values of the others. To capture this dependency, we perform a multivariate first-order Taylor expansion around a prior coupled state $\Delta_{i*}^{(0)} = \{\Delta_{ij}^{(0)}\}_{j=1}^B$, where all increments are nonzero. In practice, we treat this prior as the current state at iteration $t$ (i.e., we let $\Delta_{i*}^{(t)} := \Delta_{i*}^{(0)}$) and analyze the local descent behavior from this reference point. This converts our optimization into an iterative process, where each step refines the current set of coupled increments. The resulting linear approximation is:

$$\mathcal{L}_i(\Delta_{i*}) \approx \mathcal{L}_i(\Delta_{i*}^{(t)}) + \sum_{j=1}^B \left[ \nabla_{\Delta_{ij}}\mathcal{L}_i(\Delta_{i*}^{(t)}) \right]^\top (\Delta_{ij} - \Delta_{ij}^{(t)}). \tag{5}$$

Crucially, this expansion preserves the softmax-induced coupling: each gradient term $\nabla_{\Delta_{ij}}\mathcal{L}_i$ is evaluated assuming that all other increments $\Delta_{ik \neq j}$ remain fixed at their nonzero states. The resulting linearized loss defines a local descent landscape for each $\Delta_{ij}$. To ensure reliable updates within this approximation, we impose a trust-region constraint that bounds the update magnitude: $\|\Delta_{ij}\| \leq \varepsilon$, which reflects our belief that meaningful corrections should occur within a localized region around $t_i$. Under this constraint, the optimal update direction for each $\Delta_{ij}$ corresponds to steepest descent along its local gradient:

$$\Delta_{ij}^{(t+1)} = \Delta_{ij}^{(t)} - \alpha_{ij}^{(t)} \cdot \frac{\nabla_{\Delta_{ij}}\mathcal{L}_i(\Delta_{i*}^{(t)})}{\|\nabla_{\Delta_{ij}}\mathcal{L}_i(\Delta_{i*}^{(t)})\|}, \tag{6}$$

where step size $\alpha_{ij}^{(t)}$ is analytically determined to ensure $\|\Delta_{ij}^{(t+1)}\| \leq \varepsilon$ and derived via the Cauchy–Schwarz inequality to project onto the trust-region boundary (see Appendix H for details). This yields the steepest descent direction of the linearized loss, evaluated at a coupled batch state $\Delta_{i*}^{(t)}$, where all increments are fixed but nonzero. However, this update rule has two main limitations: (i) it only guarantees descent within the current batch, lacking cross-batch generalization; and (ii) it suffers from scale ambiguity, as the optimal trust-region radius $\varepsilon$ should vary across pairs with semantic difficulty and training stage. To overcome these issues, we adopt a learnable network $\psi$ that directly predicts the coupled increment state $\Delta_{i*}^{(t)}$ from the semantic gap $v_j - t_i$ (or $t_i - v_j$), thereby establishing cross-batch coupling among increments. The subsequent update to $\Delta_{i*}^{(t+1)}$ is implicitly performed via backpropagation on the InfoNCE loss. Since the loss gradient naturally aligns with the steepest descent direction, this formulation amortizes the iterative update process into a trainable prediction problem, enabling structure-aware and generalizable optimization over the full set of pairwise increments.

## 2.2 Gap-Aware Increment Modeling via Pair-Specific $\Delta_{ij}$

To make this optimization tractable, we replace the explicit iterative updates with a learnable function $\psi$ that directly predicts each increment $\Delta_{ij}^{(t)}$. Specifically, we amortize the descent process by training a parameterized network $\psi$ to generate the current coupled increment state based on a pairwise semantic difference and a modality-dependent context:

$$\Delta_{ij}^{(t)} = \psi\left(\eta(v_j - t_i), \mathbf{C}\,;\,\Theta^{(t)}\right), \tag{7}$$

where $\eta \in \{-1, +1\}$ controls whether the increment encodes video-specific or text-specific residual semantics. Here, $t_i$ denotes the [CLS] embedding of the text sequence, and $v_j$ is the mean-pooled representation of the frame features $\mathbf{V}_{\text{frame}}$ of the $j$-th video. We implement $\psi$ as a Cross-Attention module, with the semantic gap $v_j - t_i$ as the Query and the feature sequence $\mathbf{V}_{\text{frame}} \in \mathbb{R}^{N_v \times D}$ (or $\mathbf{T}_{\text{word}} \in \mathbb{R}^{N_w \times D}$) as the context $\mathbf{C}$ (i.e., the Key and Value). Intuitively, the query encodes what semantics are present in $v_j$ but missing in $t_i$ (or vice versa); sequence features carrying such semantics are assigned higher weights, so that the aggregated increment $\Delta_{ij}$ acts as a semantic patch correcting $t_i$. Rather than computing explicit descent steps of Eq. (6), $\psi$ learns to output increments that approximate the coupled descent direction while incorporating structure-aware priors from the context. Since $\Delta_{ij}$ directly enters the InfoNCE loss, $\psi$ is optimized end-to-end via backpropagation, enabling its outputs to align with the true gradient field. We now consider the case where $\mathbf{C} = \mathbf{V}_{\text{frame}}$ and the increment $\Delta_{ij}$ is added to the text side. In this setting, the two variables $t_i$ and $\Delta_{ij}$ share the same gradient flow: $\nabla_{t_i} \mathcal{L}_i = \sum_{j=1}^{B} \nabla_{t_{\Delta_{ij}}} \mathcal{L}_i$ and $\nabla_{\Delta_{ij}} \mathcal{L}_i = \nabla_{t_{\Delta_{ij}}} \mathcal{L}_i$, where the gradient with respect to each perturbed anchor is given by:

$$\nabla_{t_{\Delta_{ij}}} \mathcal{L}_i = \frac{(p_{ij} - y_{ij})}{\tau} \cdot \left[\frac{v_j}{\|t_{\Delta_{ij}}\| \cdot \|v_j\|} - \cos(t_{\Delta_{ij}}, v_j) \cdot \frac{t_{\Delta_{ij}}}{\|t_{\Delta_{ij}}\|^2}\right]. \tag{8}$$

This formulation introduces a gradient redistribution effect: unlike standard InfoNCE where all gradients act directly on the shared anchor $t_i$, our pair-specific design allows each negative video $v_j$ to influence only its associated increment $\Delta_{ij}$. The positive pair $(t_i, v_i)$ contributes attraction gradients to both $t_i$ and $\Delta_{ii}$, facilitating alignment; meanwhile, each negative pair $(t_i, v_j), j \neq i$, applies repulsion primarily to $\Delta_{ij}$. This leads to two key benefits: 1) tension relief: $\Delta_{ij}$ can absorb the gradient from $(t_i, v_j)$, reducing the burden on $t_i$ to decrease loss in a single step; 2) false-negative suppression: repulsion from semantically similar negatives is redirected into their respective increments, reducing the semantic bias of the anchor representation. We apply this strategy under a symmetric InfoNCE loss:

$$\mathcal{L}_{\text{info}} = -\frac{1}{2}\left(\underbrace{\frac{1}{B}\sum_{i=1}^{B}\log\frac{e^{s_{ii}/\tau}}{\sum_{j=1}^{B}e^{s_{ij}/\tau}}}_{\text{text to video}} + \underbrace{\frac{1}{B}\sum_{j=1}^{B}\log\frac{e^{s_{jj}/\tau}}{\sum_{i=1}^{B}e^{s_{ij}/\tau}}}_{\text{video to text}}\right). \tag{9}$$

This loss function maximizes the similarity of positive pairs $s(t_i + \Delta_{ii}, v_i)$ and minimizes the similarity of negative pairs.

**Norm-Based Regularization of Trust-Region Radii.** In our formulation, the increment $\Delta_{ij}$ predicted by $\psi$ is directly constrained within a trust region, thus its norm $\varepsilon_{ij} = \|\Delta_{ij}\|_2$ serves as the trust-region radius. This radius controls how far the corrected representation $t_{\Delta_{ij}}$ is allowed to deviate from $t_i$ in order to release optimization tension and adjust semantics. Intuitively, semantically similar pairs should yield smaller radii, while dissimilar pairs should allow larger ones. To encourage such structured variability, we regularize the intra-anchor distribution of radii by promoting norm diversity:

$$\mathcal{L}_{\varepsilon} = -\mathbb{E}_{t_i \sim \mathcal{B}_t}\left[\text{Var}\left(\{\varepsilon_{ij}\}_{j=1}^{B}\right)\right], \tag{10}$$

where $\mathcal{B}_t$ denotes the batch of text anchors. A lower bound $\max(\mathcal{L}_{\varepsilon}, -\lambda)$ with $\lambda > 0$ is applied to prevent instability. This regularization sharpens the implicit trust-region structure learned by $\psi$, guiding $\Delta_{ij}$ to reflect pairwise semantic variability in a stable manner.

**Directional Diversity Regularization.** To enhance the expressiveness of the learned increments $\Delta_{ij}$, we introduce a directional regularization that encourages the directions of $\{\Delta_{ij}\}_{j=1}^{B}$ under each anchor $t_i$ to be diverse. This helps the model assign distinct update directions to different candidate videos, improving the generalization of representations and mitigating mode collapse. Specifically, we normalize each increment to obtain unit vectors $z_{ij} = \frac{\Delta_{ij}}{\|\Delta_{ij}\|_2}$ and define the regularization loss as the expected angular similarity across all anchor-specific direction sets:

$$\mathcal{L}_{\mathrm{dir}} = \mathbb{E}_{t_i \sim \mathcal{B}_t} \Big[ \log \mathbb{E}_{j,k} \left[ \exp\left(-\alpha \cdot (1 - \langle z_{ij}, z_{ik} \rangle)\right) \right] \Big], \tag{11}$$

where $\alpha$ is a scale factor to control uniformity. This loss softly penalizes directional concentration, while still allowing nearby directions for semantically similar negatives—preserving flexibility under uncertainty. Combined with the norm-based regularization, this term enables fine-grained control over both the magnitude and direction of each increment $\Delta_{ij}$, leading to more stable and structure-aware optimization.

## 2.3 Variational Information Bottleneck (VIB) for Semantic Increments

We motivate our regularization from the Information Bottleneck (IB) principle [41, 1], which seeks to maximize predictive information while suppressing nuisance factors. In our case, the increment $\Delta$ is optimized only by the gradient from its paired sample $(t_i, v_j)$ and thus lacks contrastive behavior, often collapsing into trivial solutions. We therefore treat $\Delta$ as an information bottleneck that extracts semantic signals from $(t_i, v_j)$ while discarding noise, effectively constraining it within a prior structure before allowing it to reduce the InfoNCE objective. Formally, this corresponds to maximizing the mutual information between $\Delta$ and the target semantics while minimizing its dependence on the input pair:

$$\max_{p(\Delta|X)} I(\Delta; Y) - \beta\, I(\Delta; X), \tag{12}$$

where $X = (t_i, v_j)$ denotes a input pair, $Y$ the label indicating whether it is a positive match, $\Delta$ the pair-specific increment (i.e, the latent variable $Z$), and $\beta$ trade-off between task term $I(\Delta; Y)$ and compression term $I(\Delta; X)$. Following the standard variational derivation (details in Appendix D), we obtain the objective

$$\mathcal{L}_{\mathrm{VIB}} := \underbrace{-\mathbb{E}_{(t,v,y)} \mathbb{E}_{\Delta \sim q_\psi(\Delta|t,v)} \big[ \log q_\theta(y|\Delta) \big]}_{\text{contrastive term } \mathcal{L}_{\mathrm{info}}} + \beta \cdot \underbrace{\mathbb{E}_{(t,v)} \big[ \mathrm{KL}\big(q_\psi(\Delta|t,v)\|r(\Delta)\big) \big]}_{\text{compression term } \mathcal{L}_{\mathrm{IB}}}, \tag{13}$$

where $q_\psi(\Delta|t,v)$ denotes the variational encoder that parameterizes the posterior of $\Delta$ given $(t,v)$, $q_\theta(y|\Delta)$ is the variational classifier instantiated as a softmax predictor, and $r(\Delta) = \mathcal{N}(0, I)$ is the Gaussian prior used in the upper-bound regularization of the latent space.

In practice, we adopt a *deterministic instantiation* where the variational distribution $q_\psi(\Delta|t,v)$ collapses to a Dirac posterior centered at the network output $\Delta_{ij}$ (i.e., $\mu = \Delta_{ij}$), without uncertainty-aware sampling. Since the Dirac measure is singular with respect to any continuous prior, the KL divergence is ill-defined. To obtain a tractable and stable regularizer, we aggregate posteriors along the text dimension, leveraging the asymmetric nature of video–text data (i.e., videos typically contain higher redundancy and correspond to multiple semantically related texts). By the convexity of $\mathrm{KL}(\cdot\|r)$ and Jensen's inequality, this yields a relaxation that also circumvents the singularity of the deterministic posterior:

$$\begin{aligned}
\mathbb{E}_{(t,v)} \big[ \mathrm{KL}\left(q_\psi(\Delta|t,v)\|r(\Delta)\right) \big] &= \mathbb{E}_v \mathbb{E}_{t|v} \big[ \mathrm{KL}\left(q_\psi(\Delta|t,v)\|r(\Delta)\right) \big] \\
&\geq \mathbb{E}_v \big[ \mathrm{KL}\big(\bar{q}_\psi(\Delta|v)\|r(\Delta)\big) \big],
\end{aligned} \tag{14}$$

where $\bar{q}_\psi(\Delta|v) := \mathbb{E}_{t|v}[\, q_\psi(\Delta|t,v)\,]$ is the aggregated increment posterior for video $v$. This relaxation preserves the information-bottleneck effect while reducing sensitivity to text-side variability. The precise relationship between this relaxed KL term and the original mutual information $I(\Delta; X)$ is derived in Appendix E. Putting the relaxation into practice, we approximate $\bar{q}_\psi(\Delta|v_j)$ with a Gaussian fitted to the set of increments $\{\Delta_{ij}\}_{i=1}^{B}$ associated with each video anchor $v_j$. This yields the following relaxed compression loss:

$$\mathcal{L}_{\mathrm{IB}}^{\mathrm{relax}} = \mathbb{E}_{v_j \sim \mathcal{B}_v} \big[ \mathrm{KL}\big(\mathcal{N}(\mu_j, \sigma_j^2)\|\mathcal{N}(0, I)\big) \big], \tag{15}$$

where $\mu_j$ and $\sigma_j^2$ denote the mean and variance of increments $\{\Delta_{ij}\}_{i=1}^{B}$ over the text batch for each video $v_j$. This relaxed KL term operationalizes the bottleneck by regularizing increments at the video level, enforcing centered and isotropic corrections while avoiding over-penalization of text-side variability. We provide the overall training objective and inference details in Appendix F.

Table 1: Comparison results on MSR-VTT dataset on Text-to-Video Retrieval and Video-to-Text Retrieval. DiCoSA [24] utilizes QB-Norm [6] for inference and is grayed out for a fair comparison. Note that T2VLA [47] is a non-CLIP method.

| Methods | Text-to-Video Retrieval | | | | | Video-to-Text Retrieval | | | | |
|---|---|---|---|---|---|---|---|---|---|---|
| | R@1↑ | R@5↑ | R@10↑ | MdR↓ | MnR↓ | R@1↑ | R@5↑ | R@10↑ | MdR↓ | MnR↓ |
| T2VLA [47] CVPR21 | 29.5 | 59.0 | 70.1 | 4.0 | - | 31.8 | 60.0 | 71.1 | 3.0 | - |
| CLIP4Clip [34] Neurocomputing22 | 44.5 | 71.4 | 81.6 | 2.0 | 15.3 | 42.7 | 70.9 | 80.6 | 2.0 | 11.6 |
| X-Pool [17] CVPR22 | 46.9 | 72.8 | 82.2 | 2.0 | 14.3 | 44.4 | 73.3 | 84.0 | 2.0 | 9.0 |
| TS2-Net [33] ECCV22 | 47.0 | 74.5 | 83.8 | 2.0 | 13.0 | 45.3 | 74.1 | 83.7 | 2.0 | 9.2 |
| EMCL-Net [22] NeurIPS22 | 46.8 | 73.1 | 83.1 | 2.0 | 12.8 | 46.5 | 73.5 | 83.5 | 2.0 | 8.8 |
| UATVR [16] ICCV23 | 47.5 | 73.9 | 83.5 | 2.0 | 12.3 | 46.9 | 73.8 | 83.8 | 2.0 | 8.6 |
| DiCoSA [24] IJCAI23 | 47.5 | 74.7 | 83.8 | 2.0 | 13.2 | 46.7 | 75.2 | 84.3 | 2.0 | 8.9 |
| ProST [30] ICCV23 | 48.2 | 74.6 | 83.4 | 2.0 | 12.4 | 46.3 | 74.2 | 83.2 | 2.0 | 8.7 |
| HBI [23] CVPR23 | 48.6 | 74.6 | 83.4 | 2.0 | 12.0 | 46.8 | 74.3 | 84.3 | 2.0 | 8.9 |
| DiffusionRet [25] ICCV23 | 49.0 | 75.2 | 82.7 | 2.0 | 12.1 | 47.7 | 73.8 | 84.5 | 2.0 | 8.8 |
| EERCF [40] AAAI24 | 47.8 | 74.1 | 84.1 | - | - | 44.7 | 74.2 | 83.9 | - | - |
| MPT [56] ACM MM24 | 48.3 | 72.0 | 81.7 | - | 14.9 | 46.5 | 74.1 | 82.6 | - | 11.8 |
| **Baseline** | 46.6 | 73.4 | 82.2 | 2.0 | 12.6 | 45.6 | 73.4 | 82.4 | 2.0 | 9.6 |
| **GARE (Ours)** | **49.1** | 74.7 | 83.6 | **2.0** | **12.0** | **48.6** | **75.3** | **85.3** | **2.0** | **8.5** |

Table 2: Comparison results on DiDeMo, ActivityNet Captions, and MSVD datasets on Text-to-Video Retrieval. Note that FROZEN [3] is a non-CLIP method.

| DiDeMo | | | | | ActivityNet Captions | | | | | MSVD | | | | |
|---|---|---|---|---|---|---|---|---|---|---|---|---|---|---|
| Methods | R@1 | R@5 | R@10 | MnR | Methods | R@1 | R@5 | R@10 | MnR | Methods | R@1 | R@5 | R@10 | MnR |
| TS2-Net | 41.8 | 71.6 | 82.0 | 14.8 | CLIP4Clip | 40.5 | 72.4 | 83.6 | 7.5 | FROZEN [3] | 33.7 | 64.7 | 76.3 | - |
| CLIP4Clip | 42.8 | 68.5 | 79.2 | 18.9 | TS2-Net | 41.0 | 73.6 | 84.5 | 8.4 | CLIP4Clip | 45.2 | 75.5 | 84.3 | 10.3 |
| DiCoSA | 45.7 | 74.6 | 83.5 | 11.7 | DiCoSA | 42.1 | 73.6 | 84.6 | 6.8 | EMCL-Net | 42.1 | 71.3 | 81.1 | 17.6 |
| DiffusionRet | 46.7 | 74.7 | 82.7 | 14.3 | MPT | 41.4 | 70.9 | 82.9 | 7.8 | UATVR | 46.0 | 76.3 | 85.1 | 10.4 |
| HBI | 46.9 | 74.9 | 82.7 | 12.1 | HBI | 42.2 | 73.0 | 84.6 | 6.6 | Diffusion | 46.6 | 75.9 | 84.1 | 15.7 |
| **Baseline** | 45.4 | 74.3 | 82.0 | 12.3 | **Baseline** | 40.2 | 72.5 | 83.6 | 7.5 | **Baseline** | 45.0 | 75.5 | 84.5 | 10.7 |
| **GARE (Ours)** | **47.6** | **75.4** | **83.1** | **12.0** | **GARE (Ours)** | **42.6** | 73.2 | **84.8** | **6.6** | **GARE (Ours)** | 46.4 | **76.1** | 84.5 | **10.6** |

# 3 Experiment

## 3.1 Experiment settings

**Datasets and Metrics.** We evaluate our method on four standard text-video retrieval benchmarks: MSR-VTT [52], DiDeMo [2], MSVD [8], and ActivityNet Captions [27]. MSR-VTT contains 10K videos with 20 captions each; we follow the 1K-A validation split. DiDeMo includes 10K videos segmented into 5-second clips, each annotated with multiple sentences. MSVD consists of 1.9K short video clips with English captions. ActivityNet Captions provides dense annotations for 20K long-form videos with multiple temporally grounded descriptions. We choose Recall at rank K={1, 5, 10} (R@K), Median Rank (MdR), and Mean Rank (MnR) to evaluate the retrieval performance.

**Implementation Details.** We adopt CLIP (ViT-B/32) [38] as the base dual-encoder, equipped with a 4-layer Temporal Transformer [42] following the CLIP vision encoder for video encoding. Following prior works [34, 17, 30, 45], we use 32-word captions and 12 video frames for MSR-VTT and MSVD, and 64-word captions with 64 frames for DiDeMo and ActivityNet Captions due to their longer video durations. We use the Adam optimizer [14] with linear warm-up, as in prior works. The learning rate is set to 1e−7 for CLIP's text and visual encoders, and 1e−4 for all other modules. We set $\beta = 0.07$, $\tau = 0.01$, $\alpha = 2$, and $\lambda = 0.5$ for MSR-VTT. All experiments use a batch size of 128. We train the model for 5 epochs on MSR-VTT, MSVD, and DiDeMo, and 10 epochs on ActivityNet Captions. All experiments are conducted on 4 to 8 GPUs including RTX 4090, A100 and V100.

## 3.2 Comparison with Other Methods

Table 1 and Table 2 shows the performance of our method across four standard text-video retrieval benchmarks. As seen, our approach consistently outperforms recent state-of-the-art methods on MSR-VTT, ActivityNet, DiDeMo and MSVD.

Table 3: Ablation on losses combination on Text-to-Video Retrieval results on MSR-VTT 1k-A. First row denotes the baseline.

| $\Delta$ | $\mathcal{L}_{\mathbf{IB}}^{\text{relax}}$ | $\mathcal{L}_{\varepsilon}$ | $\mathcal{L}_{\mathbf{dir}}$ | R@1↑ | R@5↑ | R@10↑ | MnR↓ |
|---|---|---|---|---|---|---|---|
| Baseline | | | | 46.6 | 73.4 | 82.2 | 12.6 |
| ✓ | | | | 47.4 | 73.8 | 82.8 | 12.4 |
| ✓ | | ✓ | | 47.2 | 73.3 | 82.2 | 12.4 |
| ✓ | | | ✓ | 47.0 | 73.1 | 82.3 | 12.6 |
| ✓ | | ✓ | ✓ | 47.4 | 73.7 | 82.8 | 12.3 |
| ✓ | ✓ | | | 48.3 | 74.2 | 83.2 | 12.4 |
| ✓ | ✓ | ✓ | ✓ | **49.1** | **74.7** | **83.6** | **12.0** |

Table 4: Ablation on Context Modality Choice of $\psi$. Text-to-video retrieval results on three datasets under different context modalities.

| Dataset | Context $\mathbf{C}$ | R@1↑ | R@5↑ | R@10↑ | MnR↓ |
|---|---|---|---|---|---|
| MSR-VTT | $\mathbf{T}_{\text{word}}$ | 47.4 | **73.5** | 82.1 | 12.9 |
| | $\mathbf{V}_{\text{frame}}$ | **49.1** | 73.3 | **82.2** | **12.4** |
| ActivityNet | $\mathbf{T}_{\text{word}}$ | **42.6** | 73.6 | **84.4** | **6.8** |
| | $\mathbf{V}_{\text{frame}}$ | 40.2 | 72.2 | 83.6 | 8.1 |
| DiDeMo | $\mathbf{T}_{\text{word}}$ | 46.5 | 74.3 | 82.6 | 12.3 |
| | $\mathbf{V}_{\text{frame}}$ | **47.6** | **75.4** | **83.1** | **12.0** |

Table 5: Ablation on the interaction mode of $\psi$ on Text-to-Video Retrieval results on MSR-VTT 1k-A. The variant removes the relative gap modeling by using $t_i$ as the query and $\mathbf{V}_{\text{frame}}$ as the key–value, producing $t'_{ij}$ and $\Delta_{ij} = v_j - t'_{ij}$. Our gap-aware design preserves pair-specific structure and yields superior alignment.

| Interaction Mode of $\psi$ | R@1↑ | R@5↑ | R@10↑ | MnR↓ |
|---|---|---|---|---|
| Query $= t_i$ (no gap) | 46.1 | 73.2 | 81.9 | 13.7 |
| Query $= v_j - t_i$ | **49.1** | **74.7** | **83.6** | **12.0** |

Table 6: Ablation on the IB prior $r(\Delta)$ on MSR-VTT 1k-A. Comparison between normalized and unnormalized $\Delta_{ij}$ distributions with different Gaussian priors.

| $\sigma$ | R@1↑ | R@5↑ | R@10↑ | MnR↓ |
|---|---|---|---|---|
| *Normalized* $\Delta$ | | | | |
| 1.0 | 47.8 | 74.5 | 82.1 | 12.9 |
| *Unnormalized* $\Delta$ | | | | |
| 0.1 | 47.7 | 73.4 | 82.2 | 12.9 |
| 1.0 | **49.1** | **74.7** | **83.6** | 12.0 |
| 10.0 | 48.1 | 74.6 | 83.5 | 12.0 |
| 100.0 | 48.6 | **74.7** | 83.2 | **11.8** |

## 3.3 Ablative Analysis

All ablation studies are conducted on the MSR-VTT 1k-A validation set. In addition to the ablations presented below, we further provide results on the lower bound coefficient $\lambda$ of $\mathcal{L}_{\varepsilon}$, the interaction between the context modality type $\mathbf{C}$ of $\psi$ and the direction indicator $\eta$, the scale factor $\alpha$ of $\mathcal{L}_{\text{dir}}$, the choice of anchor in $\mathcal{L}_{\text{IB}}^{\text{relax}}$ and hard negative methods comparison in Appendix C.

**Losses Combination.** We conduct ablation studies on the MSR-VTT 1k-A validation set to assess the effectiveness of the proposed increment $\Delta$ and its associated regularizers. As shown in the right of Table 3, directly injecting $\Delta$ into the InfoNCE flow improves performance from 46.6 to 47.4, validating the benefit of gradient tension release via pairwise adjustment. Introducing the relaxed information bottleneck (IB) loss further boosts performance to 48.3, highlighting its role in guiding $\Delta$ toward semantically meaningful corrections. In contrast, adding the norm constraint or the directional diversity regularizer alone yields no gain, since each $\Delta_{ij}$ is only optimized with respect to its corresponding sample pair and thus lacks inherent contrastive behavior; simply minimizing InfoNCE can drive $\Delta$ toward trivial or collapsed solutions. The IB loss imposes a prior structure by restricting the optimization freedom of $\Delta$, and within this semantically grounded structure, the norm and diversity regularizers become truly effective; without such semantic constraints, applying them to trivial solutions of $\Delta$ would be meaningless. When combined, relaxed IB with norm and diversity regularization achieves the best performance (49.1), demonstrating that semantic grounding and structured regularization must work together to fully exploit the potential of $\Delta$.

**Impact of Cross-attention Interaction Design.** As shown in Table 5, we examine the effect of replacing $\psi$'s pair-wise gap-aware interaction with a simplified query–key setting, where $t_i$ serves as the query and $\mathbf{V}_{\text{frame}}$ as the key–value. This variant yields a fused representation $t'_{ij}$ and a residual $\Delta_{ij} = v_j - t'_{ij}$, thereby removing explicit gap modeling between $t_i$ and $v_j$. Although it captures frame-level semantics aligned with $t_i$, directly updating $t_i$ via $\Delta_{ij}$ leads to severe performance degradation, as gradients near the loss side propagate to $t_{\Delta_{ij}} = t_i + v_j - t'_{ij}$, disturbing the anchor semantics. To mitigate this, we normalize $\Delta_{ij}$ to a unit direction $\Delta_{\text{norm}}$ and compute a similarity-based magnitude $\mathcal{R}$ from frame-wise similarities between $t_i$ and $v_j$, obtained through a linear projection followed by exponentiation. The final correction $t_{\Delta_{ij}} = t_i + \mathcal{R} \cdot \Delta_{\text{norm}}$ partially alleviates gradient interference but still lacks explicit semantic-gap awareness, preventing $\psi$ from leveraging CLIP's

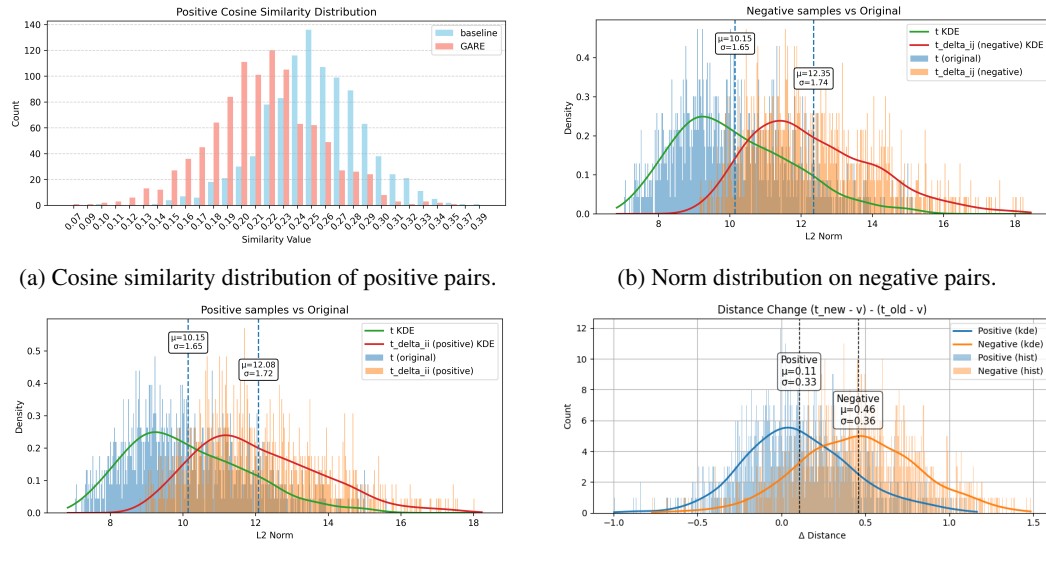

(a) Cosine similarity distribution of positive pairs.

(b) Norm distribution on negative pairs.

(c) Norms distribution on positive pairs.

(d) Distance shift: $t_\Delta$ vs. $t$.

Figure 3: Qualitative analysis on the MSR-VTT 1k-A validation set. $t_{\text{delta}}$ denotes $t_\Delta$. Our method induces greater angular separation between positive pairs (a), redistributes $t_\Delta$ norms to release gradient tension (b, c), and pushes $t_\Delta$ outward from $v_j$ (d), promoting uniformity.

prior alignment—confirming that explicit pair-wise gap modeling is crucial for robust cross-modal alignment.

**Prior $r(\Delta)$ of Information Bottleneck.** We further examine the effect of the prior distribution in the information bottleneck objective. A standard Gaussian prior is commonly used to regularize *normalized* embeddings, implicitly encouraging an isotropic distribution concentrated around a hyperspherical shell in high-dimensional space. In contrast, our best performance is achieved when the *unnormalized* $\Delta_{ij}$ distribution is regularized against the same standard Gaussian prior. As shown in Table 6, normalizing $\Delta_{ij}$ before KL regularization accelerates convergence but reduces final accuracy, as it constrains all increments to the unit sphere and removes one degree of freedom—preventing each $\Delta_{ij}$ from being regularized along its own radial axis. Moreover, since $\Delta_{ij}$ operates in the *unnormalized space* to correct the anchor $t_i$, preserving magnitude information is crucial for effective alignment. We further tested priors $r(\Delta) = \mathcal{N}(0, \sigma^2 I)$ with different standard deviation $\sigma \in \{0.1, 10, 100\}$ under the unnormalized setting, but none outperformed the standard Gaussian ($\sigma = 1$), suggesting that the optimal prior variance is data-dependent, while the standard Gaussian provides a balanced regularization strength in our case.

**Context Modality Choice in $\psi$.** We evaluate the effect of applying $\Delta_{ij}$ to either modality across datasets with contrasting characteristics. As shown in the left of Table 4, on MSR-VTT, injecting $\Delta$ on the text side (i.e., $t_i + \Delta_{ij}$) achieves the best performance, while applying it to the video side degrades results. This aligns with the fact that MSR-VTT contains many visually similar short clips, making it more effective to adjust the concise text representation for fine-grained distinctions. Conversely, on ActivityNet, applying $\Delta$ to the video side leads to a notable performance boost, whereas modifying text harms results (R@1 $\approx$ 40). This is likely because the long but redundant videos are paired with rich, structured captions—making video-side adaptation more beneficial. These trends highlight the importance of aligning $\psi$'s modality choice with dataset structure.

### 3.4 Qualitative Analysis

**Geometric Properties of $\Delta$.** To understand the effect of the learned increment $\Delta$, we conduct a qualitative analysis on the MSR-VTT 1k-A validation set, focusing on its impact on representation geometry and alignment behavior at inference. As shown in Figure 3a, our method GARE yields

lower cosine similarities between positive pairs than the baseline, indicating larger angular separation and improved uniformity on the unit hypersphere [46].

Figures 3b and 3c show that the norm of the adjusted text embedding $t_\Delta$ consistently exceeds that of the original $t$, implying that $\Delta$ expands text representations onto a series of spheres of larger radius. Positive embeddings $t_{\Delta_{ii}}$ also have greater norms, consistent with our analysis of $\Delta$'s iterative update behavior. Performing the first-order Taylor expansion around nonzero $\Delta$ states (i.e., with nonzero initialization per iteration) mitigates logit-ranking distortion that occurs when expanding solely at zero. As negative increments $\Delta_{ij}$ ($j \neq i$) are pushed outward, the positive $\Delta_{ii}$ also expands to preserve relative belief masses [39] among logits, lowering overall cosine similarity while maintaining relative softmax probabilities.

Figure 3d further shows that $t_{\Delta_{ij}}$ lies farther from $v_j$ than $t_i$, implying that the model does not simply reduce inter-modal distance but expands the text representation onto a larger spherical shell for finer alignment. This suggests that, in cosine-based contrastive learning, encouraging greater dispersion of samples in the unnormalized feature space (i.e., higher pre-projection uniformity) may facilitate more effective alignment on the unit hypersphere.

Overall, these results indicate that optimization tension is effectively released: representation learning is no longer confined to the narrow region induced by the modality gap but occurs within a broader geometric space, thereby raising the upper bound of achievable alignment performance. Additional training-time analysis is provided in Appendix G.

**Gradient Analysis.** To further analyze how $\Delta$ redistributes optimization tension during training, we visualize the per-dimension gradient statistics of $t_{\Delta_{ij}}$ at a representative training step (Figure 4). In dimensions with significant optimization activity, both positive and negative gradients reach magnitudes around $g \approx 2.5$ and appear as approximate opposites, corresponding to the pair-specific gradient form in Eq. (8).

When aggregated over all pairs, these opposite gradients largely cancel out in the anchor's update $\nabla_{t_i} \mathcal{L}_i$, leading to the near-zero gradient state described earlier. However, unlike $t_i$, which aggregates signals from all pairs, each $\Delta_{ij}$ only receives the gradient transmitted from its corresponding pair. Consequently, the positive increment $\Delta_{ii}$ receives an effective gradient of approximately $+g$, while each negative $\Delta_{ij}$ receives around $-g/B$, where $B$ denotes the batch size. Since these gradients act independently and are not mutually canceled, the total non-vanishing optimization strength per anchor $t_i$ is approximately $|+g| + B \cdot |-g/B| \approx 2|g|$, indicating that $\Delta$ components remain actively optimized rather than stagnant.

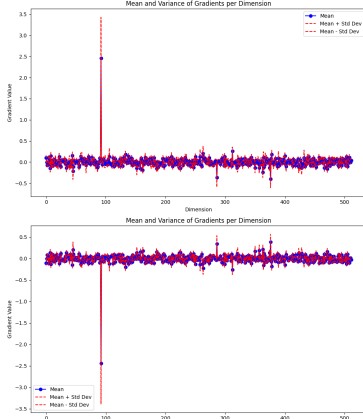

Figure 4: Mean and variance of per-dimension gradients, indicating the positive gradients (top) acting on $t_{\Delta_{ii}}$ and $\Delta_{ii}$ and the sum of all negative gradients (bottom) for $t_{\Delta_{ij}}$ and $\Delta_{ij}$.

This reveals that the trajectory of $\Delta_{ij}$ reflects how $t_i$ explores the representation space. By distributing gradients across $\Delta$, our framework effectively offloads optimization tension from the anchor, expanding its reachable region and breaking the locality constraint imposed by the modality gap.

## 4 Conclusion

We revisited contrastive learning for text–video retrieval from an optimization perspective, identifying two key challenges: optimization tension from the modality gap and gradient noise from false negatives. Through a first-order Taylor expansion of the InfoNCE loss under a trust-region constraint, we derived a function space of increments $\Delta_{ij}$ that approximate descent directions. A learnable gap-aware network predicts $\Delta_{ij}$ to redistribute gradients across pairs, expanding the optimization range beyond the modality gap and mitigating false-negative noise. To ensure $\Delta_{ij}$ encodes compact yet informative corrections, we employ a variational information bottleneck with a relaxed compression objective for stable training. Experiments on four benchmarks demonstrate consistent improvements, and further analyses show that the learned increments form structured, semantically meaningful geometry that releases optimization tension and enhances alignment capacity.

# 5 Acknowlegements

This work was supported by the NSFC NO. 62172138 and No. 62202139. This work was also partially supported by the Fundamental Research Funds for the Central Universities NO. JZ2024HGTG0310 and No. JZ2025HGTB0226.

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

# Appendix

This appendix provides additional discussions, derivations, and analyses that complement the main paper. We first discuss the limitations of our current framework and provide related works for broader context. Then, we present extended ablation studies and comparisons, followed by detailed mathematical derivations, including the relaxed form of our variational information bottleneck (VIB) objective and the rationale for using a non-zero state in the multivariate Taylor expansion. Finally, we provide implementation and efficiency details of the GARE model to support reproducibility and deployment.

## A   Limitations

While our method successfully reduces the tension between text and video embeddings by optimizing pair-wise $\Delta_{ij}$ based on the direction that minimizes the InfoNCE loss, several limitations remain:

**Lack of modality alignment.**   The fundamental issue of modality gap persists. Despite reducing the tension between the embeddings of text and video, the two modalities still reside in completely disjoint regions of the representation space. Our approach alleviates this problem by releasing the optimization tension exerted on anchor representations, thereby expanding the effective optimization range of each text anchor through multiple pair-specific increments $\Delta_{ij}$. As a result, the method mitigates the effects of the modality gap and reduces some noise, but it does not address the root cause of the misalignment between the two modalities.

**Lack of generalized supervision for $\Delta_{ij}$.**   Our approach relies heavily on adjusting $\Delta_{ij}$ through gradient-based optimization, but $\Delta_{ij}$ lacks a more generalizable supervision signal. Currently, we are optimizing each pair-wise $\Delta_{ij}$ based solely on the gradient of the InfoNCE loss, which only loosely guides the optimization direction. While this helps alleviate the tension between positive and negative pairs, it does not provide a stronger supervisory signal to guide the model toward a better generalization across unseen data.

## B   Related Work

**Contrastive learning and modality gap.**   Contrastive learning has become a foundational paradigm in multimodal representation learning. Wang and Isola [46] formalize contrastive learning via the principles of alignment and uniformity on the hypersphere, offering a geometric perspective on representation quality. Wang and Liu [44] show that contrastive loss is hardness-aware and temperature-sensitive, and reveal a trade-off between representation uniformity and semantic tolerance, highlighting the need to preserve meaningful structure among semantically similar samples. Liang et al.[31] investigate the modality gap in multimodal contrastive learning and attribute it to initialization imbalance and cone effects. False negatives have also been recognized as a key challenge in contrastive learning [12, 9], with solutions ranging from reweighting and elimination to dynamic detection and correction.

**Text-video retrieval.**   Text-video retrieval is one of the prominent tasks [15, 26, 43, 4, 18, 51, 36, 5, 24, 29, 48] in cross-modal learning. The majority of existing research [30, 16, 25, 35, 49, 7, 53] in this area utilizes a mapping technique that aligns both text and video inputs within a shared latent space to facilitate direct similarity assessment. CLIP4Clip [34] is the first to adapt CLIP [38] for video-text retrieval via temporal frame aggregation. TS2-Net [33] improves temporal modeling through token shift and selection. X-Pool [17] introduces text-guided pooling to highlight salient video tokens. HBI [22] values possible correspondences between frames and words using Banzhaf interaction for sensitive and explainable cross-modal alignment. EMCL [22] introduces an expectation-maximization [13] framework to learn a compact latent space where video and text features are represented as linear combinations of shared bases. This decomposition reduces the rank of the latent space, mitigating the modality gap and enhancing semantic alignment. Unlike prior works that refine matching structures, our method analyzes the gradient form of InfoNCE and introduces a learnable gap-aware increment $\Delta_{ij}$ to offload optimization tension, enabling structured optimization in a trust-region-aware formulation.

# C  More Ablation and Comparison Experiments

In the next four experiments conducted on the MSR-VTT 1K-A dataset, we investigated the design of $\psi$ network inputs, the impact of the scale coefficient $\alpha$ and the orthogonality variant in the directional diversity regularization, the impact of the lower bound $\lambda$ of Norm-Based Regularization of Trust-Region Radii, and the impact of the anchor choice of $\mathcal{L}_{\text{IB}}^{\text{relax}}$. We also provide hard negative methods comparison.

## C.1  Ablation on the Design of $\psi$ Network Inputs

We analyze the design of the $\psi$ network from three complementary aspects: 1) the context used for conditioning, 2) the semantic gap direction $\eta$, and 3) the corrected anchor (either $t$ or $v$). When correcting the text anchor ($t$), using the video-side sequence ($v$) as context yields better performance; conversely, when correcting the video anchor ($v$), using the text-side sequence ($t$) as context performs better. In both cases, the semantic gap is defined as $v - t$, suggesting that the query direction is asymmetric. Overall, performance is mainly determined by the interplay between the corrected anchor and the chosen context: cross-modal conditioning (e.g., correcting $t$ with $v$ as context) consistently outperforms uni-modal configurations, highlighting the benefit of cross-modal fusion.

Moreover, defining the semantic gap as $v - t$ achieves better results than $t - v$. This can be attributed to the fact that video features tend to capture more shared and general concepts, while text features encode instance-specific information. Removing text-specific components from the video representation thus encourages more generalized and robust semantic alignment.

Table 7: Ablation on context $\mathbf{C}$ and semantic gap $\eta$ under different correction modes. $\eta{=}1$ denotes $v{-}t$.

| $\mathbf{C}$ | $\eta$ | R@1↑ | R@5↑ | R@10↑ | MnR↓ |
|---|---|---|---|---|---|
| *t-corrected* | | | | | |
| $\mathbf{T}_{\text{word}}$ | $-1$ | 46.6 | 74.2 | 83.8 | 12.9 |
| $\mathbf{T}_{\text{word}}$ | $+1$ | 48.2 | 74.2 | 83.2 | 12.9 |
| $\mathbf{V}_{\text{frame}}$ | $-1$ | 47.8 | 74.1 | 82.8 | 12.4 |
| $\mathbf{V}_{\text{frame}}$ | $+1$ | **49.1** | **74.7** | 83.6 | **12.0** |
| *v-corrected* | | | | | |
| $\mathbf{V}_{\text{frame}}$ | $-1$ | 47.7 | 73.2 | 82.9 | 13.1 |
| $\mathbf{V}_{\text{frame}}$ | $+1$ | 47.5 | 73.5 | 81.8 | 13.0 |
| $\mathbf{T}_{\text{word}}$ | $-1$ | 48.2 | **73.6** | 82.8 | **12.7** |
| $\mathbf{T}_{\text{word}}$ | $+1$ | **48.8** | 73.2 | **83.6** | 12.9 |

## C.2  Ablation of the Scale Coefficient $\alpha$ and the Orthogonality Variant in Directional Diversity

To further analyze the impact of the scale coefficient $\alpha$ in the directional diversity regularization, we conduct an ablation study as shown in Table 8. Recall that the loss adopts a log-mean-exp form:

$$\mathcal{L}_{\text{dir}} = \mathbb{E}_{t_i \sim \mathcal{B}_t} \Big[ \log \mathbb{E}_{j,k} \left[ \exp\left( -\alpha \cdot (1 - \langle z_{ij}, z_{ik} \rangle) \right) \right] \Big],$$

where $z_{ij} = \frac{\Delta_{ij}}{|\Delta_{ij}|_2}$ denotes normalized directions under the same anchor $t_i$. The term $(1 - \langle z_{ij}, z_{ik} \rangle)$ lies in $[0, 2]$, and the log-mean-exp operator favors pairs with higher cosine similarity (i.e., smaller angular distance), encouraging local angular diversity without enforcing strict orthogonality.

Table 8: Ablation on scale coefficient $\alpha$ for directional diversity.

| $\alpha$ | R@1↑ | R@5↑ | R@10↑ | MnR↓ |
|---|---|---|---|---|
| 0.5 | 47.6 | 74.5 | 83.4 | 12.1 |
| 1.0 | 48.5 | 74.4 | **83.9** | 12.1 |
| 2.0 | **49.1** | **74.7** | 83.6 | **12.0** |
| 3.0 | 46.2 | 74.4 | 82.9 | 12.1 |
| 4.0 | 47.0 | 74.3 | 83.4 | 12.1 |
| 5.0 | 47.2 | 74.1 | 82.8 | 12.4 |

The coefficient $\alpha$ adjusts the strength of uniformity: 1) When $\alpha$ is small, the exponential term emphasizes pairs with higher similarity, focusing optimization on local clusters of $\Delta_{ij}$ directions. 2) As $\alpha$ increases, the optimization becomes more uniform across pairs, driving the distribution of $\Delta_{ij}$ directions toward isotropy.

Empirically, as shown in Table 8, performance peaks when $\alpha = 2.0$, achieving the best R@1, R@5 and MnR. When $\alpha$ grows larger, R@1 and R@5 exhibit reduced variance, indicating that the model becomes less sensitive to angular differences. This aligns with our interpretation that large $\alpha$ values saturate the exponential term—causing $\exp(-\alpha(1 - \langle z_{ij}, z_{ik} \rangle))$ to approach zero uniformly—thus diminishing the discriminative effect among diverse directions. Overall, a moderate $\alpha$ achieves the best trade-off between directional diversity and training stability, effectively preventing directional collapse while maintaining meaningful variation across $\Delta_{ij}$.

To further enhance the distinction among $\Delta_{ij}$ directions, we additionally experimented with a strict orthogonalization objective:

$$\mathcal{L}_{\text{orth}} = \frac{1}{B^2} \sum_{j}^{B} \sum_{k,k \neq j}^{B} \langle z_{ij}, z_{ik} \rangle^2,$$

which enforces all direction pairs under each anchor $t_i$ to be mutually orthogonal. Unlike the log-mean-exp formulation, this loss treats both high- and low-similarity pairs equally and aims to drive all pairwise cosine similarities toward zero. However, this uniform orthogonality constraint yields inferior performance (R@1=47.7, R@5=74.3, R@10=83.1, MnR=12.1). We attribute this degradation to the excessive suppression of negatively correlated directions—pairs that are semantically opposite or far apart are also pushed toward zero similarity.

This rigid treatment prevents $\Delta_{ij}$ from encoding opposing semantic relations, indicating that directional diversity should remain semantically flexible rather than uniformly orthogonal.

Nevertheless, the orthogonalization approach exhibits stronger performance on video-to-text retrieval (R@1=49.4, R@5=75.4, R@10=83.6, Mean R=8.2). We conjecture that this improvement stems from the fact that the directional diversity regularization is applied with text as the anchor over candidate videos, where enforcing orthogonality among $\Delta_{ij}$ encourages the model to better separate diverse visual counterparts for the same textual query.

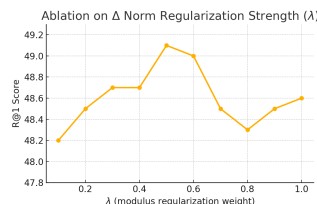

Figure 5: R@1 score with varying $\lambda$ for $\Delta$ norm regularization.

## C.3 Ablation on $\Delta$ Norm Regularization Strength

We investigate the effect of varying the lower bound factor $\lambda$, which controls the target margin of $\varepsilon$ separation within each anchor $t_i$. As shown in Figure 5, increasing $\lambda$ from 0.1 to 0.5 improves R@1, with the best performance at $\lambda = 0.5$. This suggests that moderate diversity in $\varepsilon_{ij}$ is beneficial for enhancing semantic discrimination. However, as $\lambda$ increases further, performance degrades, likely due to over-regularization.

To mitigate the instability introduced by hard truncation (i.e., directly thresholding $\varepsilon_{ij}$), we also experiment with a smooth approximation using a log-sum-exp formulation:

$$\mathcal{L}_{\varepsilon-\text{LSE}} = \log \left[ 1 + \frac{1}{B} \sum_{j=1}^{B} \exp \left[ -(\varepsilon_{ij} - \bar{\varepsilon}_i)^2 \right] \right].$$

Although this variant imposes a natural lower bound and provides vanishing gradients near convergence, it does not significantly improve retrieval. We hypothesize that this is due to gradient saturation when the variance among $\varepsilon_{ij}$ becomes too large, which in turn weakens the ability to further enforce directional separation.

## C.4 Ablation on the Anchor Choice in Relaxed VIB Regularization

We further investigate the effect of the anchor choice in the relaxed information bottleneck term $\mathcal{L}_{\text{IB}}^{\text{relax}}$. In the main formulation, the KL regularization is anchored on videos, i.e.,

$$\mathbb{E}_{(t,v)}[\text{KL}(q_\psi(\Delta|t,v)\|r(\Delta))] \geq \mathbb{E}_v\left[\text{KL}(\bar{q}_\psi(\Delta|v)\|r(\Delta))\right],$$

where $\bar{q}_\psi(\Delta|v) = \mathbb{E}_{t|v}[q_\psi(\Delta|t,v)]$. This relaxation aggregates increments $\Delta_{ij}{}_{i=1}^{B}$ under each video anchor $v_j$ and preserves the information-bottleneck effect while reducing sensitivity to the variability of short text inputs.

Table 9: Effect of anchor choice in KL regularization $\mathcal{L}_{\text{IB}}^{\text{relax}}$. Anchoring on $v$ yields better performance.

| Anchor | R@1 | R@5 | R@10 | MdR | MnR |
|---|---|---|---|---|---|
| $t$ | 47.9 | **74.9** | **83.8** | **2.0** | 12.5 |
| $v$ | **49.1** | 74.7 | 83.6 | **2.0** | **12.0** |

To contrast this design, we also test the reversed relaxation that anchors on text:

$$\mathbb{E}_{(t,v)}[\text{KL}(q_\psi(\Delta \mid t, v) \| r(\Delta))] \geq \mathbb{E}_t\left[\text{KL}(\bar{q}_\psi(\Delta \mid t) \| r(\Delta))\right],$$

where $\bar{q}_\psi(\Delta|t) := \mathbb{E}_{v|t}[q_\psi(\Delta|t,v)]$. Practically, this means that the set $\{\Delta_{ij}\}_{j=1}^{B}$ associated with each text anchor $t_i$ is modeled as a Gaussian posterior, yielding the corresponding compression loss.

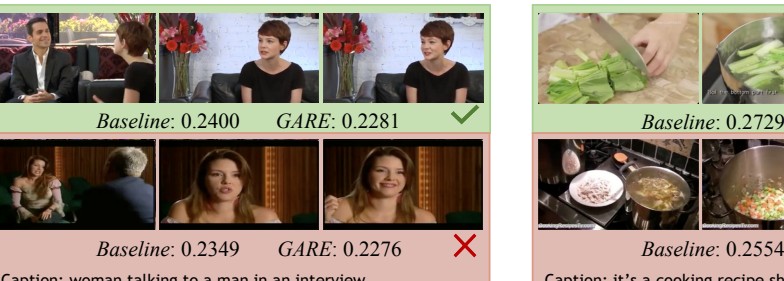

Query: a woman on a couch talks to a man

*Baseline*: 0.2400    *GARE*: 0.2281  ✓

*Baseline*: 0.2349    *GARE*: 0.2276  ✗

Caption: woman talking to a man in an interview

Query: a person is putting the vegetable in to the water and boil it

*Baseline*: 0.2729    *GARE*: 0.2488  ✓

*Baseline*: 0.2554    *GARE*: 0.2410  ✗

Caption: it's a cooking recipe show with chicken vegetables

(a)                                                    (b)

Figure 6: Comparison of hard negative alignment before and after applying $\Delta_{ij}$ optimization. Compared with the baseline, GARE produces smaller similarity gaps among semantically related videos $v_j$. This indicates that GARE effectively mitigates the noise from hard negatives and reduces the semantic deviation of the anchor $t_i$, leading to more stable and consistent alignment across similar samples.

Notably, our relaxation introduces stochasticity directly from the dataset pairing itself, rather than by sampling additional uncertainty variables.

As shown in Table 9, anchoring the KL term on the video side leads to superior retrieval performance (R@1=49.1 vs. 47.9). We attribute this to the inherent data characteristics: the textual descriptions in the dataset are typically short and less informative, while videos are semantically richer and often correspond to multiple captions. Consequently, conditioning $\Delta$ on $v$ captures more shared, modality-invariant structure, aligning better with the underlying multimodal distribution and yielding stronger retrieval results.

## C.5 Hard Negative Comparison

We compare **GARE** with two recent methods that explicitly or implicitly handle hard negatives:

**DMAE** [21] (R@1: 46.9, +1.6% over base. *ACM MM 2023*): DMAE enhances fine-grained alignment by mining hard positives—for instance, text queries associated with specific frames—which implicitly improves the model's capability to separate hard negatives. Conceptually, this shares similarity with our variational information bottleneck (IB) loss, where $\Delta$ is compressed through a bottleneck to retain critical alignment signals while filtering noisy gradients.

**NeighborRetr** [32] (R@1: 49.5, +2.3% over base. *CVPR 2025*): This method employs a memory bank to compute $k$-neighbor co-occurrence statistics, identifying "good hubs" that encourage local consistency and reduce over-penalization of hard negatives. Although not explicitly formulated as hard-negative mining, the top-$k$ co-occurrence selection serves a similar role by adaptively reweighting difficult negatives.

**GARE** (R@1: 49.1, +2.5% over base): Unlike the above methods, GARE does *not* rely on explicit mining or external memory structures. Each increment $\Delta_{ij}$ absorbs loss gradients locally from its paired $(t_i, v_j)$ sample, mitigating the reliance on global contrastive comparisons when encountering hard negatives. This local reallocation of gradients softens noisy updates and improves generalization. As illustrated in Figure 6, the text anchor $t_i$ in GARE exhibits enhanced semantic stability: for semantically related videos $v_j$, the similarity values $s_{ii}$ and $s_{ij}$ vary smoothly, indicating that $\Delta_{ij}$ enables more accurate fine-grained alignment across similar samples.

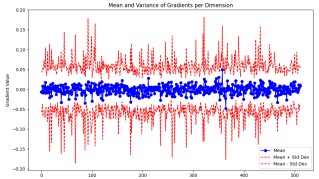

Figure 7: Mean and variance of total gradients acting on $t_i$ on each dimension.

This aligns with the gradient visualization in Figure 7, where the gradients on $t_i$'s dimensions remain centered around zero after introducing $\Delta$, demonstrating its role as a stable semantic prototype.

**Efficiency comparison.** GARE maintains superior efficiency despite comparable or higher performance. It uses only a single cross-attention layer, whereas NeighborRetr includes 8 MLP modules and

multiple transformer or convolutional blocks. NeighborRetr also requires large-scale memory banks (10,240 samples per modality) and ~4.5 h training time, while GARE achieves similar accuracy with 1 h 34 min of training and negligible additional memory cost compared to the baseline. Together, these results demonstrate that GARE achieves comparable hard-negative robustness with substantially lower computational overhead.

# D  Derivation of the VIB Objective for Pair-Specific Increments

We model each pair-specific increment as a latent variable $Z = \Delta_{ij}$ for the input $X = (t_i, v_j)$ and the match label $Y \in \{0, 1\}$. Our objective maximizes predictive information under a compression constraint:

$$\max_{p(z|x)} I(Z; Y) - \beta\, I(Z; X), \quad \beta > 0. \tag{16}$$

Below we derive a tractable variational lower bound of Eq. (16).

**Lower bound for $I(\Delta_{ij}; Y)$.**  For convenience, in the following, we will refer to $y_{ij}$, $\Delta_{ij}$, $t_i$, and $v_j$ as $y$, $\Delta$, $t$, and $v$ respectively. We make the standard assumption that the match label does not depend on $\Delta$ once $(t, v)$ is given:

$$p(y|\Delta, t, v) \;=\; p(y|t, v). \tag{17}$$

By definition,

$$
\begin{aligned}
I(\Delta; y) &= \iint p(\Delta, y) \log \frac{p(y|\Delta)}{p(y)} \, d\Delta dy \\
&= \iint p(\Delta, y) \log \Big( \frac{q_\theta(y|\Delta)}{p(y)} \cdot \frac{p(y|\Delta)}{q_\theta(y|\Delta)} \Big) \, d\Delta dy \\
&= \iint p(\Delta, y) \log \frac{q_\theta(y|\Delta)}{p(y)} \, d\Delta dy + \mathrm{KL}\big(p(y|\Delta)\|q_\theta(y|\Delta)\big) \\
&\geq \iint p(\Delta, y) \log \frac{q_\theta(y|\Delta)}{p(y)} \, d\Delta dy \quad \text{(by non-negativity of KL)} \\
&= \iint p(\Delta, y) \log q_\theta(y|\Delta) \, d\Delta dy + H(Y) \\
&\geq \iiint p(y, \Delta, t, v) \log q_\theta(y|\Delta) \, d\Delta dy d(t, v) \\
&= \iiint p(y|t, v) p(\Delta|t, v) p(t, v) \log q_\theta(y|\Delta) \, d\Delta dy d(t, v) \\
&= \mathbb{E}_{(t, v, y)} \mathbb{E}_{\Delta \sim q_\psi(\Delta|t, v)} \big[ \log q_\theta(y|\Delta) \big].
\end{aligned}
\tag{18}
$$

where $q_\theta(y|\Delta)$ is a variational classifier, the non-negativivty entropy term $H(Y)$ is constant w.r.t. model parameters, and the last second equality uses assumption Eq. (17). Let $q_\psi(\Delta|t, v)$ be the variational encoder. Then,

$$I(\Delta; y) \geq \mathbb{E}_{(t, v, y)} \, \mathbb{E}_{\Delta \sim q_\psi(\Delta|t, v)} \big[ \log q_\theta(y|\Delta) \big]. \tag{19}$$

The two inequalities above arise from the non-negativity of $\mathrm{KL}(\cdot\|\cdot)$ and from rewriting $\log \frac{1}{p(y)}$ as the entropy $H(y)$.

**Upper bound for $I(\Delta_{ij}; t_i, v_j)$.** Using $I(\Delta; t, v) \leq \mathbb{E}_{(t,v)}\mathrm{KL}(p(\Delta|t,v)\|r(\Delta))$ with any auxiliary prior $r(z)$,

$$
\begin{aligned}
I(\Delta; t, v) &= \iint p(\Delta, t, v) \log \frac{p(\Delta, t, v)}{p(\Delta)\, p(t, v)}\, d\Delta d(t, v) \\
&= \iint p(\Delta, t, v) \log \frac{p(\Delta|t, v)}{p(\Delta)}\, d\Delta d(t, v) \\
&= \iint p(\Delta, t, v) \log \left( \frac{p(\Delta|t, v)}{r(\Delta)} \cdot \frac{r(\Delta)}{p(\Delta)} \right) d\Delta d(t, v) \\
&= \iint p(\Delta, t, v) \log \frac{p(\Delta|t, v)}{r(\Delta)}\, d\Delta d(t, v) - \mathbb{E}_{(t,v)}\mathrm{KL}(p(\Delta)\|r(\Delta)) \\
&\leq \iint p(\Delta, t, v) \log \frac{p(\Delta|t, v)}{r(\Delta)}\, d\Delta d(t, v) \\
&= \mathbb{E}_{(t,v)}\big[\mathrm{KL}(p(\Delta|t, v)\|r(\Delta))\big]
\end{aligned}
\tag{20}
$$

where the inequality follows from $\mathrm{KL}(p(\Delta)\|r(\Delta)) \geq 0$ and $p(\Delta|t, v)$ denotes the variational encoder $q_\psi(\Delta|t, v)$ that parameterizes the model's conditional distribution of the pair-specific increment (deterministic in our implementation). We use a diagonal Gaussian prior $r(z) = \mathcal{N}(0, I)$.

**VIB lower bound.** Combining Eq. (18) and Eq. (20) yields the variational lower bound of Eq. (16), i.e., the upper bound of the negative of Eq. (16):

$$
\mathcal{L}_{\mathrm{VIB}} := -\mathbb{E}_{(t,v,y)}\mathbb{E}_{\Delta \sim q_\psi(\Delta|t,v)}\big[\log q_\theta(y|\Delta)\big] + \beta \cdot \mathbb{E}_{(t,v)}\big[\mathrm{KL}(q_\psi(\Delta|t, v)\|r(\Delta))\big].
\tag{21}
$$

# E  Relation between the Relaxed VIB Compression Term and the Original IB Objective

We start from the standard information bottleneck (IB) formulation, which regularizes the mutual information between the learned increment $\Delta$ and the input pair $(t, v)$:

$$
I(\Delta; T, V) = \mathbb{E}_{p(t,v)}[\mathrm{KL}(q_\psi(\Delta \mid t, v) \,\|\, q_\psi(\Delta))].
\tag{22}
$$

Since the marginal $q_\psi(\Delta)$ is intractable, it is commonly replaced by a fixed prior $r(\Delta)$, leading to the following upper bound:

$$
\mathbb{E}_{(t,v)}[\mathrm{KL}(q_\psi(\Delta \mid t, v) \,\|\, r(\Delta))] = I(\Delta; T, V) + \mathrm{KL}(q_\psi(\Delta) \,\|\, r(\Delta)).
\tag{23}
$$

This term penalizes the total amount of information $\Delta$ retains about both modalities and serves as the compression loss in the original VIB objective.

**Relaxation anchored on the video side.** To mitigate sensitivity to text-side variability, we anchor the expectation on videos and aggregate increments over all textual pairs associated with the same $v$. By convexity of the KL divergence (Jensen's inequality), we have:

$$
\begin{aligned}
\mathbb{E}_{(t,v)}[\mathrm{KL}(q_\psi(\Delta \mid t, v) \,\|\, r(\Delta))] &= \mathbb{E}_v \mathbb{E}_{t|v}[\mathrm{KL}(q_\psi(\Delta \mid t, v) \,\|\, r(\Delta))] \\
&\geq \mathbb{E}_v[\mathrm{KL}(\bar{q}_\psi(\Delta \mid v) \,\|\, r(\Delta))],
\end{aligned}
\tag{24}
$$

where the aggregated posterior $\bar{q}_\psi(\Delta \mid v) = \mathbb{E}_{t|v}[q_\psi(\Delta \mid t, v)]$ represents the mixture distribution of increments under the same video anchor.

Applying the decomposition in Eq. (23) to the right-hand side of Eq. (24) yields

$$
\mathbb{E}_v[\mathrm{KL}(\bar{q}_\psi(\Delta \mid v) \,\|\, r(\Delta))] = I_{\bar{q}}(\Delta; V) + \mathrm{KL}(q_\psi(\Delta) \,\|\, r(\Delta)),
\tag{25}
$$

where $I_{\bar{q}}(\Delta; V)$ denotes the mutual information defined under the aggregated distribution $\bar{q}_\psi$. Combining the two equations gives

$$
I(\Delta; T, V) \;\geq\; I_{\bar{q}}(\Delta; V),
\tag{26}
$$

indicating that the relaxation effectively removes the conditional term $I(\Delta; T \mid V)$ from the full IB regularization.

**Interpretation.** The relaxed KL term therefore optimizes

$$\mathcal{L}_{\text{IB}}^{\text{relax}} = \mathbb{E}_v[\text{KL}(\bar{q}_\psi(\Delta \mid v) \,\|\, r(\Delta))] = I_{\bar{q}}(\Delta; V) + \text{KL}(q_\psi(\Delta) \,\|\, r(\Delta)), \tag{27}$$

which serves as a lower bound of the original compression term. While the standard IB loss penalizes all information in $\Delta$ about $(T, V)$, the relaxed form only constrains information shared with $V$ and ignores the conditional mutual information $I(\Delta; T \mid V)$. Intuitively, this relaxation preserves the bottleneck effect but allows $\Delta$ to encode text-specific variations within each video cluster, yielding more stable optimization when multiple captions correspond to the same visual content.

# F    Overall Objective and Inference

The overall training objective is formulated as

$$\mathcal{L}_{\text{total}} = \underbrace{\mathcal{L}_{\text{info}} + \beta \, \mathcal{L}_{\text{IB}}^{\text{relax}}}_{\mathcal{L}_{\text{VIB}}} + \lambda_\varepsilon \, \mathcal{L}_\varepsilon + \lambda_{\text{dir}} \, \mathcal{L}_{\text{dir}}, \tag{28}$$

where the first two terms constitute the VIB optimization objective, and $\lambda_\varepsilon$ and $\lambda_{\text{dir}}$ are the weights of the structural regularizers. In practice, we set $\beta = 0.07$ and $\lambda_\varepsilon = \lambda_{\text{dir}} = 0.01$ for MSR-VTT. During inference, we retain the learned increment $\Delta$ to assist retrieval, as it encodes the semantic consistency between $t_i$ and $v_j$. We further discuss the efficiency of using $\Delta$ at both training and inference stages in the Appendix I, as well as its distinction from traditional dual-encoder retrieval paradigms and deployment strategies for large-scale retrieval.

# G    Analysis of $\Delta$ Behavior During Training

To better understand the role and dynamics of the learned semantic-gap vector $\Delta$ throughout training, we visualize four key aspects of $\Delta$'s behavior, presented in Figure 8a to 8d. These analyses reveal the underlying geometric transformations and provide further insight into how $\Delta$ facilitates optimization under InfoNCE loss. We will continue conducting ablation studies on the Text-to-Video retrieval task using the MSR-VTT [52] dataset.

**Angle between $\Delta$ and $(v_j - t_i)$.** As shown in Figure 8a, we track the angle between $\Delta$ and the initial cross-modal gap vector $v_j - t_i$ throughout training, for both positive and negative pairs. At initialization, this angle is close to $\frac{\pi}{2}$, indicating that $\Delta$ is nearly orthogonal to $v_j - t_i$, and thus carries no meaningful alignment with the cross-modal semantic gap. This implies that the early $\Delta$ vectors do not differentiate between positive and negative pairs, acting more like isotropic perturbations in space. As training proceeds, however, this angle gradually increases into the obtuse region for both positive and negative samples. This reflects a significant shift in behavior—rather than attempting to directly bridge the semantic gap $v_j - t_i$, the model learns to push $t_i$ away from $v_j$, effectively offloading the gradient tension induced by the modality gap and false negative interference. This offloading allows contrastive learning to take place in an expanded embedding space, where $\Delta$ modulates the representation geometry to ease the optimization burden. This trend is corroborated by Figure 8d, where the Euclidean distances between the updated text embeddings $t_{\Delta_{ij}} = t_i + \Delta_{ij}$ and $v_j$ become significantly larger than the original distances $\|t_i - v_j\|$, for both positive and negative samples. That is, $\Delta$ introduces a global scaling effect in the representation space, and contrastive optimization is carried out under a larger geometric regime.

Interestingly, the new positive pair distances $\|t_{\Delta_{ii}} - v_i\|$ are also larger than their original counterparts $\|t_i - v_i\|$. Though this seems counterintuitive—since the gradient of InfoNCE with respect to $\Delta$ pushes toward $v_i$ — it actually reflects the relative nature of the InfoNCE loss. The network does not aim to minimize absolute distances, but rather to increase the similarity margin between matched and mismatched pairs. Thus, pushing all embeddings outward in norm (as further validated in Figure 8c) gives the model more room to maneuver in angular space, while the cosine similarity objective remains stable under such rescaling. This aligns with our design intent: to leverage $\Delta$ as a structural carrier for modality-aware tension redistribution, providing optimization flexibility on a normalized manifold.

Moreover, in early training stages, the angles between $\Delta$ and $v_j - t_i$ are nearly identical for positive and negative samples, showing that the model has not yet learned to encode fine-grained pairwise

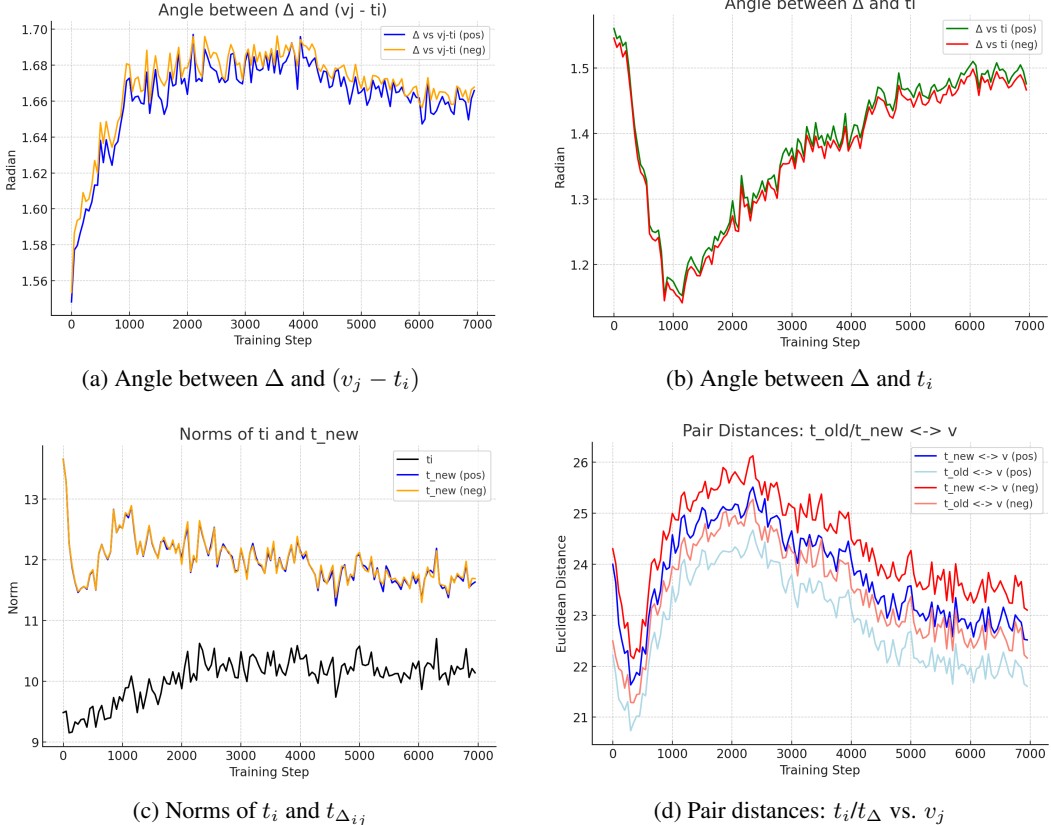

(a) Angle between $\Delta$ and $(v_j - t_i)$

(b) Angle between $\Delta$ and $t_i$

(c) Norms of $t_i$ and $t_{\Delta_{ij}}$

(d) Pair distances: $t_i/t_\Delta$ vs. $v_j$

Figure 8: Training dynamics of the learned modality-gap vector $\Delta$. $t_{\text{new}}$ denotes $t_\Delta$. (a) $\Delta$ grows increasingly orthogonal to the initial modality gap, indicating embedding space expansion. (b) $\Delta$ initially aligns with the anchor $t_i$, then deviates to encode semantic distinctions. (c) Updated embeddings $t_{\text{new}}$ operate under larger norms, enlarging the contrastive space. (d) Pairwise distances increase and then stabilize, reflecting semantic separation and convergence.

differences. But as training advances, a slight but consistent gap emerges—$\Delta$ for positive samples tends to have slightly smaller angles than that of negatives. This subtle divergence signals that $\Delta$ has begun to capture semantically meaningful pair-level distinctions, enabling more discriminative alignment in later stages.

**Angle between $\Delta$ and $t_i$.**     As shown in Figure 8b, the angle between $\Delta$ and $t_i$ exhibits a distinct pattern: it decreases rapidly in the early training phase, indicating that $\Delta$ is initially aligned with the anchor text embedding. This suggests that the model's default behavior is to trust the prior structure of the text space and apply similar $\Delta$ directions across different $v_j$, especially when pairwise semantic differences are not yet learned.

However, over time, this alignment loosens—$\Delta$ deviates further from $t_i$ as the model begins to adapt to modality-specific differences. This marks a transition where the network $\psi$ no longer treats the anchor $t_i$ as a default direction and instead generates $\Delta$ based on nuanced distinctions across the visual features $v_j$, thus leading to more expressive and semantically grounded $\Delta$ vectors.

**Pair distances between text and video features.**     Figure 8d visualizes the Euclidean distances between text and video pairs over the course of training, comparing both the original pairs $t_i \leftrightarrow v_j$ and the updated pairs $t_{\Delta_{ij}} \leftrightarrow v_j$, across positive and negative samples. Several important patterns emerge: First, we observe that positive pair distances remain consistently smaller than negative pair distances, both in the original and updated forms. This confirms that the learned $\Delta$ not only preserves the basic alignment structure of contrastive learning but also enhances semantic discrimination, effectively encoding fine-grained pairwise structure through the transformation. Second, for both positive and

negative pairs, the distances between $t_\Delta$ and $v$ are larger than those between the original $t$ and $v$, demonstrating that the model pushes the updated text embeddings into a larger-scale embedding regime. This is precisely in line with our gradient tension offloading design: $\Delta$ introduces a controlled displacement that alleviates the direct optimization pressure on $t_i$, allowing contrastive comparisons to operate within a higher-norm, more expressive space, as also supported by Figure . The evolution of the distance curves over time also reflects meaningful training dynamics. Initially, all distances decrease, which we interpret as a normalization phase — embedding distributions at initialization are noisy and unstructured, and the model first compresses them into a tighter, more consistent geometric configuration. This is followed by a steady increase in distances, as the model begins to explicitly separate positive and negative samples to satisfy the InfoNCE objective. Finally, all curves converge and stabilize into a bounded range, suggesting that the semantic configuration of the embedding space has reached a relatively converged structural state.

Taken together, these observations highlight the critical role of $\Delta$ not only in scaling the embedding geometry but also in encoding structurally-aware, semantically-aligned displacements that facilitate tension redistribution and enable robust representation learning.

## H  Why We Don't Use an Initial $\Delta_{i*} = 0$ Expansion and the Need for a Non-Zero Initial $\Delta_{i*}$

### H.1  Introduction to the Issue

In the previous approach, we considered using multiple variables $\Delta_{ij}$ to distribute the gradient impact of each video $v_j$ on the text embedding $t_i$. However, the specific form of $\Delta_{ij}$ was unclear. The idea was to update each $\Delta_{ij}$ in the direction that would reduce the InfoNCE loss. A natural first step was to expand the loss function $\mathcal{L}_i$ as a Taylor expansion around the initial state $\Delta_{i*} = 0$, which corresponds to an expansion solely based on $t_i$.

However, this expansion introduces a key issue: the gradient terms in each $\Delta_{ij}$ expansion are computed under the assumption that all other $\Delta_{ik}$ remain zero. This neglects the mutual interactions among $\{\Delta_{ij}\}_{j=1}^{B}$ and decouples the updates for different pairs. Since the InfoNCE loss fundamentally depends on the *relative ranking* of cosine similarities between $t_i$ and all candidate videos $\{v_j\}$, such a decoupled formulation disrupts the prior ordering structure already established by the CLIP encoder's logits. In other words, it breaks the model's inherent relational prior among candidates, which is essential for maintaining consistent contrastive comparisons.

### H.2  Why Expanding with $\Delta_{i*} = 0$ Is Inadequate

When $\Delta_{ij}^* = 0$ for all $j$, the gradient expansion for each pair $\Delta_{ij}$ looks like:

$$\mathcal{L}_i(\Delta_{ij}) \approx \mathcal{L}_i(0) + \nabla_{\Delta_{ij}} \mathcal{L}_i(0)^\top \Delta_{ij}, \tag{29}$$

where $\nabla_{\Delta_{ij}} \mathcal{L}_i(0)$ is the gradient of the loss evaluated at $\Delta_{ij} = 0$. In this case, the gradient $p_{ij} - y_{ij}$ is computed as:

$$\nabla_{\Delta_{ij}} \mathcal{L}_i(0) = (p_{ij}(0) - y_{ij}) \nabla_{\Delta_{ij}} \cos_{ij}(0), \tag{30}$$

where $p_{ij}(0)$ is the softmax term evaluated at the cosine similarity $\cos(t_i, v_j)$. This expansion does not account for the interdependence between different $\Delta_{ik}$, and the gradients are computed as if each $\Delta_{ij}$ were independent of all other $\Delta_{ik}$, which violates the relative comparison required for contrastive learning.

### H.3  The Need for a Non-Zero Initial $\Delta_{i*}$

To avoid this issue, we recognize that each $\Delta_{ij}$ should be updated in a way that reflects the interdependence between different pairs, not just based on an independent text-video pair. The gradient of the InfoNCE loss must be evaluated with respect to the current state of all $\Delta_{ij}$, not just the initial zero state.

Thus, we need to initialize $\Delta_{i*}$ in a way that reflects the current state of all other pairs rather than assuming they are zero. The update for each $\Delta_{ij}$ should respect the relationship between all the pairwise similarities, as the optimization of one pair affects the relative similarity between all other pairs.

In this case, the correct approach is to perform the first-order Taylor expansion around a non-zero state $\Delta_{ij}^{(t)}$ that has already been optimized for several steps. This allows us to incorporate the effect of each pair $\Delta_{ij}$ on the entire set of pairwise comparisons, thus preserving the relative relationships that are critical for InfoNCE loss.

### H.4  Optimization Objective and Trust-Region Constrained Solution

Having established the need for a coupled, non-zero initialization of $\Delta_{ij}$ to preserve the relative structure required by InfoNCE, we now formulate the optimization problem for finding the approximate update direction under a trust region constraint. Specifically, we assume a trust region radius $\varepsilon$ that bounds the magnitude of each $\Delta_{ij}$ and analyze the solutions in both the single-step (non-iterative) and iterative update settings.

**Non-iterative first-order update.**  We begin by considering the first-order Taylor expansion of the per-anchor loss $\mathcal{L}_i$ with respect to a single variable $\Delta_{ij}$ evaluated at the origin:

$$\mathcal{L}_i(\Delta_{ij}) \approx \mathcal{L}_i(0) + \nabla_{\Delta_{ij}}\mathcal{L}_i(0)^\top \Delta_{ij}. \tag{31}$$

Since the constant term $\mathcal{L}_i(0)$ does not affect optimization, minimizing the approximated loss is equivalent to minimizing the linear term under a trust-region constraint:

$$\min_{\|\Delta_{ij}\| \leq \varepsilon}  \nabla_{\Delta_{ij}}\mathcal{L}_i(0)^\top \Delta_{ij}, \tag{32}$$

where $\varepsilon$ denotes the radius limiting the update magnitude of $\Delta_{ij}$. By the Cauchy–Schwarz inequality, for $g_{ij} = \nabla_{\Delta_{ij}}\mathcal{L}_i(0)$ we have

$$g_{ij}^\top \Delta_{ij} \geq -\|g_{ij}\| \|\Delta_{ij}\| \geq -\|g_{ij}\| \varepsilon, \tag{33}$$

which provides a tight lower bound for the objective within the trust region. The bound is attainable when $\Delta_{ij}$ is colinear with $-g_{ij}$ and satisfies $\|\Delta_{ij}\| = \varepsilon$, yielding the feasible update

$$\Delta_{ij}^* = -\varepsilon \cdot \frac{g_{ij}}{\|g_{ij}\|}. \tag{34}$$

This closed-form solution achieves the maximal decrease in the first-order approximation of $\mathcal{L}_i$, corresponding to the steepest descent direction constrained by the trust region. However, this derivation treats $\mathcal{L}_i$ as a function of a single $\Delta_{ij}$ and neglects the coupling among $\{\Delta_{ik}\}_{k \neq j}$, thus breaking the relative similarity structure essential to InfoNCE. A more faithful formulation must therefore incorporate all $\Delta_{ij}$ jointly, as discussed in the following subsection.

**Iterative update with coupled expansion.**  We next analyze the update rule when $\Delta_{ij}$ is expanded around a coupled, non-zero state $\Delta_{i*}^{(t)} = \{\Delta_{ij}^{(t)}\}_{j=1}^{B}$, where each gradient $\nabla_{\Delta_{ij}}\mathcal{L}_i$ is computed under the influence of all other nonzero $\Delta_{ik}^{(t)}$. We seek the next update step within a trust region:

$$\Delta_{i*}^{(t+1)} = \Delta_{i*}^{(t)} + \delta, \quad \|\Delta_{i*}^{(t+1)}\| \leq \varepsilon. \tag{35}$$

The first-order Taylor expansion of the per-anchor loss $\mathcal{L}_i$ at this coupled state is:

$$\mathcal{L}_i(\Delta_{i*}^{(t+1)}) \approx \mathcal{L}_i(\Delta_{i*}^{(t)}) + \sum_{j=1}^{B} \left[\nabla_{\Delta_{ij}}\mathcal{L}_i(\Delta_{i*}^{(t)})\right]^\top \left(\Delta_{ij}^{(t+1)} - \Delta_{ij}^{(t)}\right). \tag{36}$$

This multivariate linearization serves to reveal that each partial gradient $\nabla_{\Delta_{ij}}\mathcal{L}_i(\Delta_{i*}^{(t)})$ is conditioned on the current coupled state, and thus inherently encodes the interactions among all pairs. We do not minimize the linearized approximation itself; rather, it clarifies that the correct descent direction for each $\Delta_{ij}$ should be taken from the true gradient of $\mathcal{L}_i$ evaluated at the coupled state.

Following the principle of steepest descent, we update each component by

$$\Delta_{ij}^{(t+1)} = \Delta_{ij}^{(t)} - \alpha_{ij}^{(t)}\hat{g}_{ij}^{(t)}, \qquad \hat{g}_{ij}^{(t)} = \frac{\nabla_{\Delta_{ij}}\mathcal{L}_i(\Delta_{i*}^{(t)})}{\|\nabla_{\Delta_{ij}}\mathcal{L}_i(\Delta_{i*}^{(t)})\|}. \tag{37}$$

To satisfy the trust-region constraint $\|\Delta_{ij}^{(t+1)}\| \leq \varepsilon$, we determine the maximal feasible step size $\alpha_{ij}^{(t)}$ from

$$\|\Delta_{ij}^{(t)} - \alpha_{ij}^{(t)}\hat{g}_{ij}^{(t)}\|^2 \leq \varepsilon^2, \tag{38}$$

which expands to the quadratic inequality:

$$\alpha_{ij}^{(t)2} - 2\alpha_{ij}^{(t)}\Delta_{ij}^{(t)\top}\hat{g}_{ij}^{(t)} + \|\Delta_{ij}^{(t)}\|^2 - \varepsilon^2 \leq 0. \tag{39}$$

The maximal feasible solution is given by:

$$\alpha_{ij}^{(t)} = \Delta_{ij}^{(t)\top}\hat{g}_{ij}^{(t)} + \sqrt{\left(\Delta_{ij}^{(t)\top}\hat{g}_{ij}^{(t)}\right)^2 - \|\Delta_{ij}^{(t)}\|^2 + \varepsilon^2}. \tag{40}$$

This update ensures that each $\Delta_{ij}$ moves along its steepest descent direction while keeping its magnitude within the trust-region radius $\varepsilon$. Importantly, because all gradients $\nabla_{\Delta_{ij}}\mathcal{L}_i(\Delta_{i*}^{(t)})$ are computed under the coupled multivariate state, the optimization preserves the relative ranking structure among all candidate pairs required by the InfoNCE objective.

## H.5 Error Analysis

The iterative update of $\Delta_{ij}$ is derived from a multivariate first-order Taylor expansion of $\mathcal{L}_i$ under a trust-region constraint. This formulation defines a *descent function space* that guarantees loss reduction under the coupled gradient field, rather than prescribing a unique update direction for each $\Delta_{ij}$. Once the trust-region radius $\varepsilon_{ij}$ is specified (assuming an $\ell_2$-ball constraint), the steepest-descent direction becomes unique, and the theoretical update moves $\Delta_{ij}$ onto the boundary where the Cauchy–Schwarz inequality reaches equality. However, in practice, we do not explicitly assign a radius $\varepsilon_{ij}$ to each pair; instead, the effective trust-region constraint emerges implicitly from the model's prior, which we regularize through a norm-based penalty that controls the magnitude of $\Delta$. Specifically, the $\Delta_{ij}$ produced by the $\psi$ network during the forward pass corresponds to the current state $\Delta_{ij}^{(t)}$, and the optimizer update during backpropagation embeds the next-step state $\Delta_{ij}^{(t+1)}$ into the parameters of $\psi$. When the model performs the next forward pass, the generated $\Delta_{ij}$ naturally reflects this updated state, while the norm-based regularization acts as the trust-region constraint on the previous update step. Therefore, the theoretical and practical procedures share the same steepest-descent direction, but their update *magnitudes* may differ due to the implicit learning of the trust-region radius through model priors. Consequently, our error analysis focuses on the consistency of update magnitudes between the theoretically derived trust-region step and the actual updates produced by the training framework.

**Analytic step length.** At the equality boundary of the trust-region constraint, the maximal feasible step size $\alpha_{ij}^{(t)}$ satisfying $\|\Delta_{ij}^{(t+1)}\| \leq \varepsilon$ is given by:

$$\alpha_{ij}^{(t)} = \Delta_{ij}^{(t)\top}\hat{g}_{ij}^{(t)} + \sqrt{\left(\Delta_{ij}^{(t)\top}\hat{g}_{ij}^{(t)}\right)^2 - \|\Delta_{ij}^{(t)}\|^2 + \varepsilon^2}, \tag{41}$$

which projects the update exactly to the boundary of the feasible region and enforces norm-bounded motion of $\Delta_{ij}$.

**Learned step length.** During training, the neural module $\psi$ is optimized end-to-end via AdamW, implicitly learning how to adjust $\Delta_{ij}$ through gradient descent. After one step, the actual update can be expressed as:

$$\Delta_{ij}^{(t+1)} = \Delta_{ij}^{(t)} - \eta_{ij}^{(t)}\frac{\nabla_{\Theta^{(t)}}\mathcal{L}_i^{(t)}}{\|\nabla_{\Theta^{(t)}}\mathcal{L}_i^{(t)}\|}, \tag{42}$$

where $\eta_{ij}^{(t)}$ denotes the *effective step size* produced by the optimizer. Although $\Delta_{ij}$ is updated through the training framework rather than by explicitly solving the analytic trust-region problem, we apply

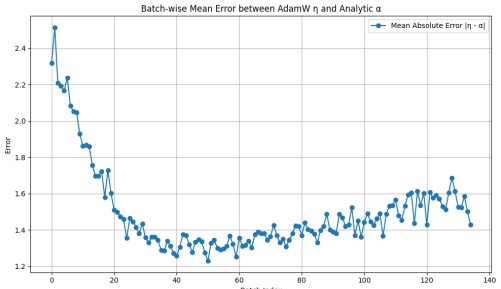
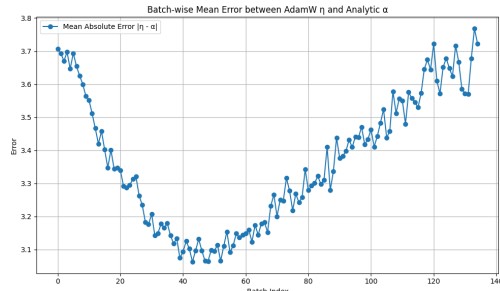

(a) Batch-wise mean error for positive pairs ($i = j$), showing convergence and bounded deviation ($[1.0, 4.5]$).

(b) Batch-wise mean error for negative pairs ($i \neq j$), exhibiting divergence that promotes embedding uniformity.

Figure 9: Comparison between the analytic step length $\alpha_{ij}$ and the effective AdamW step length $\eta_{ij}$. Although $\Delta_{ij}$ is learned through backpropagation, the norm-based trust-region constraint ensures that its magnitude remains bounded and consistent with the analytic formulation. Positive pairs show convergence and bounded errors, while negative pairs exhibit divergence that aids global uniformity.

Table 10: Comparison of computational efficiency between CLIP4Clip and GARE.

| Metric | CLIP4Clip | GARE |
|---|---|---|
| Training time | 1h 30m | 1h 34m |
| Inference time | 7.6s | 6.9s |
| Training memory (reserved) | 4×12175 MB | 4×12561 MB |
| Inference memory (reserved) | 4136 MB | 4216 MB |
| Training FLOPs (per batch) | 39,167.58 GFLOPs | 39,287.55 GFLOPs (+0.3%) |
| Inference FLOPs (per batch) | 11,868.70 GFLOPs | 11,905.05 GFLOPs (+0.3%) |

a norm-based trust-region regularization $\mathcal{L}_\varepsilon$ that restricts $\|\Delta_{ij}\|$ within a trust-region radius. This guarantees that the update magnitudes remain consistent with the theoretical constraint, even when learned implicitly.

To measure the deviation between the theoretical and learned step magnitudes, we compute the scalar error:

$$E_{ij}^{(t)} = \|\eta_{ij}^{(t)} - \alpha_{ij}^{(t)}\|. \tag{43}$$

This metric quantifies how closely the effective update magnitude in training adheres to the analytic trust-region step size.

**Empirical analysis.** Figure 9a and Figure 9b visualize the batch-wise mean error for positive and negative pairs, respectively. For positive pairs ($i = j$), the error $\|\eta_{ii} - \alpha_{ii}\|$ remains within a stable range of $[1.0, 4.5]$ and shows a clear convergence trend. Given the 512-dimensional embedding space, this corresponds to a per-dimension deviation of roughly $4.5/512$, which is negligible. This indicates that the learned $\Delta_{ii}$ updates remain consistent with the analytic trust-region dynamics and are well regularized by the norm constraint.

In contrast, for negative pairs ($i \neq j$), the error $|\eta_{ij} - \alpha_{ij}|$ shows a divergent trend, which is theoretically expected. Each negative $\Delta_{ij}$ is influenced by repulsive gradients with $(512 - 1)$ degrees of freedom, while the positive pair has a single dominant alignment direction. Moreover, since our norm-based regularization enlarges the updates whose magnitudes exceed the batch-wise mean $\|\Delta_{i*}\|$, the dispersion of negative $\Delta_{ij}$ naturally increases. Such divergence, however, plays a constructive role: by expanding negative $\{\Delta_{ij}\}, (j \neq i)$ into a larger representational space, the model enhances embedding uniformity, allowing positive $\Delta_{ii}$ to achieve more stable alignment under a broader geometric margin.

**Summary.** In summary, our analysis demonstrates that while $\Delta_{ij}$ is optimized implicitly through the training framework, the norm-based trust-region constraint effectively maintains the magnitude of $\Delta$ within the theoretical bound. The learned $\psi$ module reproduces the analytic trust-region

Table 11: Module-wise GFLOPs breakdown during forward propagation.

| Module | GFLOPs (%) | Main computation source |
|---|---|---|
| CLIP visual encoder | 11,673.60 (98.05%) | ViT image processing |
| Video temporal transformer | 38.81 (0.33%) | FFN + self-attention |
| $\psi$ (cross-attention) | 36.35 (0.31%) | linear projection |
| CLIP text encoder | 156.29 (1.31%) | text sequence processing |
| Total | 11,905.05 (100%) | |

---

**Inference Procedure (Pseudocode)**

\# Precompute all text and video embeddings offline using CLIP encoders

```
for each text batch T:
  for each video batch V:
      # Compute pair-specific delta via ψ and adjust text embeddings
      logits = model.get_similarity_logits(T, V)
      # Append cosine-similarity block to the global similarity matrix
      sim_matrix.append(logits)
      # Discard the delta tensor immediately (transient variable)
```
\# Concatenate all similarity blocks for final retrieval

---

Figure 10: Inference pseudocode of GARE. $\Delta$ are computed on-the-fly and discarded immediately to ensure constant memory usage during large-scale retrieval.

---

**Cross-Attention Pairwise Parallelization (Simplified Code)**

```
def cross_attention(query, key, value):
  # query:  (a, b, dim) for (v_j - t_i), a is text batch size, b is video
batch size
  # key/value:  (b, f, dim) for video frames

  query = query.permute(1,2,0)  # (a,b,dim) -> (b,dim,a)
  Q = Q_proj(query)
  K = K_proj(key)
  V = V_proj(value)
  # This op is pairwise-parallelizabble cross all pairs in the batch
  logits = matmul(K, Q)  # (b,f,dim) x (b,dim,a) -> (b,f,a)
  scores = softmax(logits / sqrt(dim), dim=frame_dim)

  scores = scores.permute(0,2,1)  # (b,f,a) -> (b,a,f)

  out = matmul(scores, V)  # (b,a,f) x (b,f,dim) -> (b,a,dim)
  out = out.permute(1,0,2)  # (b,a,dim) -> (a,b,dim)

  return out
```

---

Figure 11: Simplified implementation of $\psi$'s cross-attention operation. The computation is pairwise-parallelizable across all text–video pairs, ensuring linear scaling with batch size and high GPU utilization.

behavior for positive pairs, while the controlled divergence of negative pairs improves embedding uniformity—both consistent with the objectives of contrastive learning.

# I   Model Efficiency and Implementation Details

GARE remains fully compatible with the standard dual-tower CLIP architecture while introducing only a lightweight cross-modal adjustment module, $\psi$. Implemented as a single-layer cross-attention transformer without FFN expansion, $\psi$ adds merely **1.58M parameters** to the **354M** parameters of the CLIP encoders. Its computational cost is negligible—only **36.35 GFLOPs** per batch compared to CLIP's **11,868.70 GFLOPs**.

During inference, $\psi$ operates after both text and video embeddings have been precomputed, applying transient, pair-specific deltas before cosine similarity is calculated. This design preserves the pre-computability and scalability of the dual-tower framework: all embeddings can be cached offline, $\psi$ performs only on-the-fly adjustments, and no delta tensors are stored—only similarity scores are retained.

Efficiency comparisons on MSR-VTT (4×RTX 4090 GPUs, batch size 128) show that GARE introduces minimal overhead (Table 10). A module-wise breakdown (Table 11) further confirms that over 98% of total FLOPs come from the visual encoder, while $\psi$ contributes less than 0.4%, demonstrating computational parity with the original CLIP.

As illustrated in Figure 10, GARE performs inference in a batch-parallel streaming manner, generating and discarding deltas on-the-fly to maintain constant memory usage. The internal cross-attention process (Figure 11) is pairwise-parallelizable, enabling simultaneous computation across all text–video pairs and ensuring high GPU utilization.

Overall, GARE retains the efficiency of dual-tower architectures while introducing a transient, pair-specific adjustment that enhances fine-grained cross-modal alignment without increasing latency or memory cost.

**Distinction from traditional dual-Tower models.** Traditional dual-tower architectures define a consistent metric space where the triangle inequality holds, e.g., for any text–video pair $(t_i, v_j)$ and another video $v_k$,

$$d(t_i, v_j) \leq d(t_i, v_k) + d(v_k, v_j).$$

This property allows efficient large-scale retrieval via approximate nearest neighbor (ANN) search, since all embeddings share a unified metric geometry.

In contrast, GARE introduces pair-specific adjustments through $\Delta_{ij}$, yielding a conditional distance

$$d_{\text{GARE}}(t_i, v_j) = d(t_i + \Delta_{ij}, , v_j),$$

which depends on the paired video $v_j$. Substituting $t_i + \Delta_{ij}$ and $t_i + \Delta_{ik}$ into the above inequality breaks the shared metric assumption, and thus the triangle inequality no longer strictly holds. Consequently, while GARE improves fine-grained alignment, it cannot directly support ANN-based retrieval relying on fixed metric consistency.

From an efficiency standpoint, GARE computes $\Delta$ in a pair-wise parallel manner within each batch, discarding them immediately after computing cosine similarities. This design keeps latency low but introduces additional FLOPs that make direct large-scale application challenging. Nevertheless, we find that in practice the retrieved candidates from the unadjusted text embeddings $t_i$ already cover most of the top-ranked results produced by GARE. For instance, on the MSR-VTT 1k-A validation set, the top-256 candidates retrieved using $t_i$ include all of GARE's top-10 matches.

Therefore, GARE naturally supports a two-stage retrieval pipeline: (1) use the dual-tower model with $t_i$ for large-scale ANN retrieval to obtain a compact candidate set, (2) re-rank the top-$k$ candidates using GARE's pair-specific adjustments. This strategy preserves the scalability of dual-tower models while benefiting from GARE's fine-grained alignment refinement.

