# OpenReview forum: "Rebalancing Contrastive Alignment with Bottlenecked Semantic Increments in Text-Video Retrieval"
_NeurIPS.cc/2025/Conference — NeurIPS 2025 poster_

### Official Review · Reviewer_SveN · 2025-06-27

**Clarity:** 2
**Significance:** 3
**Originality:** 2
**Rating:** 4
**Confidence:** 3

**Summary:**

This paper presents GARE, a video-text retrieval framework that adjusts each pair's similarity by a learnable increment $\Delta_{ij}$. $\Delta_{ij}$ can be imposed on the embedding of any modality to reduce the modality gap and mitigate the impact of false negatives. Experiments on four popular text-video retrieval datasets show the effectiveness of GARE.

**Questions:**

Please see weaknesses.

**Ethical Concerns:**

["NO or VERY MINOR ethics concerns only"]

**Final Justification:**

I think this is a technically good paper, with its main weakness being clarity. Since the authors have committed to adding the missing technical details in the revised manuscript, I have decided to increase my rating.

**Limitations:**

Yes.

**Paper Formatting Concerns:**

Not found.

**Quality:**

3

**Strengths And Weaknesses:**

Strengths:

- The studied problem, i.e., modality gap and false negatives, is practical in cross-modal retrieval.
- The proposed method that refines the embedding by an increment is novel and instructive.
- The paper presents good qualitative analysis.

Weaknesses:
- This paper is not easy to follow. Many technical details are unclear. ①Is the final training goal the sum of various loss functions? Or does it need to be adjusted through trade-off parameters? ②What is the model structure of $\psi$? And how does it handle input from both the modality gap and the context vector?
- In general, the modality gap is a global metric to measure inherent differences, i.e., $|| \bar{v} -\bar{t}||$. But the proposed method tackles it in a pair-specific way, which requires some reasonable explanation.
- Missing related works in contrastive learning/cross-modal retrieval with false negatives.
- Compared methods are not strong, and many of them are worse than the basic CLIP model. The proposed method should be compared with noise-robust (designed for false negatives) methods.
- Does the learned network $\psi$ also generalize to inference time to refine the representation?

---

> ### Author Rebuttal · Authors · 2025-07-30
>
> Thank you very much for your positive feedback on our method. We also apologize for the unclear explanations in the manuscript that caused difficulties in reading. We will address your questions one by one.
>
> ---
>
> # **Q1: Clarification on Technical Details**
>
> - **On the training objective**: We use weight parameters **$\beta$, $\lambda_{\varepsilon}$, and $\lambda_{\text{dir}}$** to adjust the overall training objective, which can be divided into two main parts: one is the **VIB optimization objective** composed of **$\mathcal{L}\_{\text{info}} + \beta \cdot \mathcal{L}\_{\text{IB}}$**, and the other is the **delta structural regularization** composed of **$\lambda_{\varepsilon} \cdot \mathcal{L}\_{\varepsilon} + \lambda\_{\text{dir}} \cdot \mathcal{L}\_{\text{dir}}$**. The full training objective is:
>
>   $\mathcal{L}\_{\text{total}} = \underbrace{\mathcal{L}\_{\text{Info}} + \beta \cdot \mathcal{L}\_{\text{IB}}}\_{\text{VIB objective}} + \lambda_{\varepsilon} \cdot \mathcal{L}\_{\varepsilon} + \lambda_{\text{dir}} \cdot \mathcal{L}\_{\text{dir}}.$
>
>   For MSR-VTT, we set **$\beta = 0.07$**, **$\lambda_{\varepsilon} = \lambda_{\text{dir}} = 0.01$**. Since the adjustments are based on the geometric properties of the non-normalized delta embeddings, the two **$\lambda$** parameters need to be kept small (around **1e-2**) to ensure training stability.
>
>  - **On the $\psi$ structure**: We apologize for the unclear explanation of $\psi$ in the manuscript. **$\psi$** is implemented as a **single-layer cross-attention Transformer**, where the cross-attention structure is combined with an FFN that does not have the 4× expansion. In our method, we use **$v_j - t_i$** as the query input, where **$t_i$** is the `[CLS]` embedding output from the CLIP text encoder (tensor shape: **(a, dim)**, where **a** is the batch size of text dimension), and **$v_j$** is the mean of **f** frames from the j-th video sample (the frame emb with the shape of **(b, f, dim)** are obtained by the 4 layers of the video temporal Transformer after CLIP vision encoder, where **b** is the batch size of video dimension). So, the input for the pair-wise computation is of shape **(a, b, dim)**.
>    - For the cross-attention part of the $\psi$ module, since we perform pair-wise delta generation, the corresponding attention score calculation is also parallelized in a pair-wise manner. Specifically, given that the query has a shape of $(a, b, \text{dim})$, we first permute it to $(b, \text{dim}, a)$, and then perform a batch matrix multiplication with the key:
> $\text{logits}=\text{key} \times \text{query} \quad \text{i.e.,} \quad (b, f, \text{dim}) \times (b, \text{dim}, a) \to (b, f, a).$
> Next, we apply softmax to the logits and permute them again to $(b, a, f)$. Finally, the attention output is computed as:
>
>      $\text{attn-out} = \text{attn-scores} \times \text{value}, \quad \text{i.e.,} \quad (b, a, f) \times (b, f, \text{dim}) \to (b, a, \text{dim}).$
>
>      This achieves pair-wise parallel computation, with virtually no delay introduced in practice.
>
> ---
>
> # **Q2: Explanation Regarding the Modality Gap**
>
> Thank you for raising the question about the modality gap. In fact, the motivation of our work is not to directly address the modality gap itself (i.e., our method does not aim to reduce $|| \overline{v} - \overline{t} ||$), but rather to resolve the optimization tension caused by the modality gap. By releasing this tension, we achieve larger $\text{Var}(t)$ or $\text{Var}(v)$, allowing us to optimize in a broader space. At the same time, in retrieval tasks, reducing the modality gap is a sufficient but not necessary condition for improving retrieval performance. Even with a large gap, we can still achieve high retrieval performance. By optimization tension, we refer to the issue that arises when optimizing an anchor $t_i$ with respect to its InfoNCE sub-loss:
>
> $\mathcal{L}\_i = -\log \frac{e^{\cos(t_i, v_i)/\tau}}{\sum_{j=1}^B e^{\cos(t_i, v_j)/\tau}}.$
>
> The gradient of this loss can be decomposed into the form:
>
> $\nabla_{t_i} \mathcal{L}\_i = \sum_{j=1}^B \underbrace{(p_{ij} - y_{ij})}\_{\nabla_{\cos(t_i,v_j)}\mathcal{L}\_i} \cdot \underbrace{\left[ \frac{v_j}{\|t_i\|\|v_j\|} - \cos(t_i, v_j) \cdot \frac{t_i}{\|t_i\|^2} \right]}\_{\nabla_{t_i}\cos(t_i, v_j)}.$
>
> where $y_{ij}$ is the matching label.
>
> In this optimization process, \$(p\_{ii}-1) \cdot \nabla\_{t\_i} \cos(t\_i, v\_i)\$ pulls \$t\_i\$ towards \$v\_i\$, while \$p\_{ij} \cdot \nabla\_{t\_i} \cos(t\_i, v\_j)\$ pushes \$t\_i\$ away from \$v\_j\$. However, the modality gap forces the optimization of \$t\_i\$ in two directions: 1) towards \$\overline{v}\$, and 2) away from \$\overline{v}\$. We collected gradients of \$t\_i\$ during training on the CLIP4Clip model, including the positive sample gradient \$(p\_{ii}-1) \cdot \nabla\_{t\_i} \cos(t\_i, v\_i)\$ and the sum of the negative sample gradients \$\sum\_{j=1, j \neq i}^B p\_{ij} \cdot \nabla\_{t\_i} \cos(t\_i, v\_j)\$. Summing these gradients and calculating the mean and variance across the 512 dimensions, we found that for the dimensions mainly involved in optimization, the positive gradient is around 40-60, while the negative gradient sum is between -40 and -60. When both of them are summed, the resulting value for these dimensions is around -3 to -4. This strongly suggests that the modality gap induces optimization tension, as the gradient error is concentrated near zero, causing stagnation in \$t\_i\$ optimization and severely limiting the model's alignment ability.
>
> *(As the rebuttal guidelines prevent the inclusion of plots, I am unable to provide the statistical analysis visuals. If the reviewer's  interested, I can provide a detailed explanation of how we observe this phenomenon in CLIP4Clip.)*
>
> To address the optimization tension on $t_i$, we reallocate the gradient to $\Delta$, where each $\Delta$ corresponds to a specific pair and only receives gradients from the corresponding pair. Essentially, this approach lifts the ceiling on cosine similarity computation because, in the presence of optimization tension, the optimization of $t_i$ is confined to a small region near it, but the introduction of $\Delta$ expands the optimization space. As shown in Figure 2 in the paper, qualitative analysis demonstrates that the introduction of $\Delta$ alleviates the tension (enabling matching in a larger scale space) while also improving the uniformity characteristics (lower similarity distribution for positive pairs).
>
> ---
>
> # **Q3&Q4: Cross-modal Retrieval and False Negatives**
>
> Thank you for your suggestion regarding the missing related works on false negatives and the comparison with methods designed to handle false negatives. The methods compared in the current paper are all based on CLIP, typically involving structural modifications and fine-tuning on the well-known CLIP4Clip model used in TVR tasks. In the revised version, we will include additional methods related to false negatives for comparison, including the following cross-modal approaches:
>
> - **CUSA** (AAAI 2024) addresses false negatives by using soft-label alignment from pre-trained models, which reduces the impact of hard labels and enhances sample recognition.
> - **VSE++** (BMVC 2018) improves performance by focusing on the hardest negatives within each batch via Max-Hinge Loss, making the model more robust to false negatives.
> - **L2RM** (CVPR 2024) utilizes an Optimal Transport framework to realign mismatched pairs, improving robustness against hard negatives by refining sample alignment.
>
> These works address false negatives through various techniques, such as soft-label alignment, hard negative mining, and improved robustness metrics.
>
> ### **Comparison with Stronger Methods Designed for False Negatives**
>
> We now compare **GARE** with two recent methods designed to handle hard negatives in text-video retrieval:
>
> - **DMAE** (ACM MM 2023, R@1: 46.9, +1.6% over base, our reproduction: 47.1) improves fine-grained alignment by mining hard positives, which implicitly helps push away hard negatives. This is conceptually similar to our **IB loss**, which uses a variational bottleneck to compress $\Delta$, retaining critical alignment signals for robust optimization.
> - **NeighborRetr** (CVPR 2025, R@1: 49.5, +2.3% over base, our reproduction: 49.2) uses a memory bank to compute k-neighbor co-occurrence and identify "good hubs," promoting local consistency and reducing over-penalization of hard negatives. However, **GARE** does not require explicit hard negative mining. Instead, when encountering hard negative sample $v_j$, it shifts part of the loss gradient responsibility to the corresponding $\Delta_{ij}$, allowing $\Delta_{ij}$ to absorb the noisy gradient acting on $t_i$. This approach helps alleviate the erroneous optimization of $t_i$, thereby mitigating the noise introduced by hard negatives.
>
> ### **Efficiency Comparison**
>
> - **GARE** uses only **one cross-attention layer**, while **NeighborRetr** includes 8 MLPs and multiple transformer/conv layers.
> - **NeighborRetr** requires **10240-sample memory banks** and about **4.5 hours of training**, whereas **GARE** achieves similar performance with **1 hour and 34 minutes** of training and minimal memory.
>
> ### **Empirical Observations**
>
> We observe that $t_i$ in GARE becomes semantically more stable: for semantically similar $v_j$, the similarities $s(t_i, v_i)$ and $s(t_i, v_j)$ are smoother than in the baseline. This supports more accurate $\Delta$-based alignment and aligns with Fig. 2, where $t_i$ functions as a stable prototype. Ultimately, the enhanced semantic stability of $t_i$ allows $\Delta$ to carry out more precise fine-grained alignment across similar samples.
>
> ---
>
> # **Q5: $\psi$ in inference stage**
>
> Sorry for not making this clear in the paper. The $\psi$ module also participates in the inference phase during forward propagation, contributing to refining the representation for consistent matching responsibility.

---

> > ### Comment · Reviewer_SveN · 2025-08-05
> >
> > Thanks for your rebuttal. I think this is a technically good paper, with its main weakness being clarity. Since the authors have committed to adding the missing technical details in the revised manuscript, I have decided to increase my rating.

---

> ### Author Response · Authors · 2025-08-05
>
> Dear Reviewer,
>
> I hope this message finds you well. As the discussion period is nearing its end with **less than three days remaining**, I wanted to ensure we have addressed all your concerns satisfactorily. If there are any additional points or feedback you'd like us to consider, please let us know. Your insights are invaluable to us, and were eager to address any remaining issues to improve our work.
>
> Thank you for your time and effort in reviewing our paper.

---

> ### Author Response · Authors · 2025-08-06
>
> Thank you for your thoughtful response and for the constructive and insightful comments.
>
> We sincerely appreciate your patience in reviewing our work and providing valuable suggestions.
>
> We will carefully incorporate the missing technical details and clarifications into the revised manuscript and appendix.
> Thank you again for your support and for helping us improve the quality of our paper.

---

### Official Review · Reviewer_3HTR · 2025-07-01

**Clarity:** 3
**Significance:** 3
**Originality:** 3
**Rating:** 4
**Confidence:** 3

**Summary:**

This paper proposes GARE, a method for improving text-video retrieval by addressing issues in contrastive learning. It introduces a learnable pair-specific adjustment that helps reduce gradient conflicts caused by modality gaps and false negatives. The approach shows consistent performance gains across multiple benchmarks.

**Questions:**

please see the weaknesses.

**Ethical Concerns:**

["NO or VERY MINOR ethics concerns only"]

**Final Justification:**

The authors' clarifications and empirical evidence have largely addressed my concerns. I will keep my original score unchanged.

**Limitations:**

yes

**Quality:**

3

**Strengths And Weaknesses:**

Pros:
1. Introduces a novel mechanism (∆_{ij}) to locally adjust representation gaps between text and video, rather than forcing global alignment.
2. Provides a solid theoretical foundation using first-order Taylor expansion under trust-region constraints.
3. Effectively addresses both modality gap and false negatives, which are common challenges in contrastive learning.

Cons:
1. While the authors derive ∆_{ij} updates from a multivariate first-order Taylor approximation under a trust-region constraint, the actual implementation relies on a learned approximation using a neural module. The paper doesn’t sufficiently quantify how closely this learned update matches the ideal gradient direction. Including some error analysis between the true and learned update directions would strengthen the claim of "structure-aware" correction.
2. The method introduces a pairwise increment ∆_{ij} for each text-video pair, which potentially scales quadratically with batch size. However, the paper does not address the memory or computational overhead introduced by this component, nor how it compares to other methods like hard negative mining or momentum encoders.
3. From the ablation results, norm and direction regularization only help when combined with the IB loss. This suggests some over-regularization may occur if these components are applied individually. The paper might benefit from a discussion of when to apply these losses and how to tune their trade-off hyperparameters.

---

> ### Author Rebuttal · Authors · 2025-07-28
>
> We sincerely thank Reviewer for the insightful and constructive comment. We apologize for the potential ambiguity in our original manuscript regarding the interpretation of the delta update mechanism.
>
> ---
>
> ## **Q1: Error Analysis**
>
> We clarify that the update rule for each $\Delta_{ij}^{(t+1)}$, derived from a multivariate first-order Taylor expansion under a trust-region constraint, is not meant as a unique or ideal direction, but rather as a principled descent direction compatible with InfoNCE optimization. The Taylor expansion offers a general update framework, not a fixed solution.
>
> To address the batch-local limitation, we introduce a neural module $\psi$ to generate $\Delta_{i*}^{(t)}$ and embed the update into end-to-end training. Backpropagation allows deltas to evolve implicitly, capturing transferable and structure-aware patterns across batches while remaining consistent with the theoretical formulation.
>
> Since the descent direction is not unique, a strict vector-wise comparison is infeasible. Instead, we assess consistency via step size, focusing on the analytic length $\alpha_{ij}^{(t)}$ that enforces the trust-region radius $\varepsilon$:
>
> $\left\| \Delta_{ij}^{(t)} - \alpha_{ij}^{(t)} \hat{g}\_{ij}^{(t)} \right\|^2 \le \varepsilon^2, \quad \text{where} \quad \hat{g}\_{ij}^{(t)} = \frac{ \nabla_{\Delta_{ij}} \mathcal{L}\_i^{(t)} }{ \left\| \nabla_{\Delta_{ij}} \mathcal{L}\_i^{(t)} \right\| }, \varepsilon=\|\Delta_{ij}^{(t+1)}\|.$
>
> We take the **equality case** to project the update precisely to the **boundary of the feasible region**, and expand the squared norm leads to a quadratic equation in $\alpha_{ij}^{(t)}$:
>
> $\alpha_{ij}^{(t)^2} - 2\alpha_{ij}^{(t)} \Delta_{ij}^{(t)\top} \hat{g}\_{ij}^{(t)} + \left\| \Delta_{ij}^{(t)} \right\|^2 - \varepsilon^2 = 0,$
>
> whose positive root gives the **closed-form solution**:
>
> $\alpha_{ij}^{(t)} = \Delta_{ij}^{(t)\top} \hat{g}\_{ij}^{(t)} + \sqrt{ \left( \Delta_{ij}^{(t)\top} \hat{g}\_{ij}^{(t)} \right)^2 - \left\| \Delta_{ij}^{(t)} \right\|^2 + \varepsilon^2 }.$
>
> In contrast, the neural module $\psi$ is updated through AdamW, and after one optimization step, the actual update can be written as:
>
> $\Delta_{ij}^{(t+1)} = \Delta_{ij}^{(t)} - \eta_{ij}^{(t)} \cdot \frac{ \nabla_{\Theta^{(t)}} \mathcal{L}\_i^{(t)} }{ \left\| \nabla_{\Theta^{(t)}} \mathcal{L}\_i^{(t)} \right\| },$
>
> where $\eta_{ij}^{(t)}$ denotes the **effective step size** resulting from the AdamW update. To quantify the closeness between the learned and analytic updates, we compute the scalar error:
>
> $\left| \eta_{ij}^{(t)} - \alpha_{ij}^{(t)} \right|.$
>
> ------
>
> **Empirical results:**
>
> - For **positive pairs** ($i = j$):
>   We observe that the error $\left| \eta_{ii} - \alpha_{ii} \right|$ remains within a bounded range of **[1.0, 4.5]** during training and shows a clear trend of **convergence**. Considering the 512-dimensional embedding space, this deviation is moderate and acceptable.
>
> - For **negative pairs** ($i \ne j$):
> the error is larger and non-convergent, reflecting their inherently repulsive role without consistent alignment supervision.
> These fluctuations are meaningful: by expanding negative $\Delta_{ij}$ norms and lowering their cosine similarity, they enhance embedding uniformity, a key property in contrastive learning.
>
> ------
>
> In summary, the scalar step-size analysis shows that $\psi$ produces updates consistent with the analytic trust-region rule for positive pairs, validating its structure-aware behavior, while the divergence on negative pairs naturally serves to promote embedding uniformity, in line with contrastive learning objectives.
>
> *(We will include the error plots in the appendix and public codebase if the paper is accepted, as current rebuttal guidelines do not permit figures.)*
>
> ---
>
> ## **Q2: Effeciency and Hard Negative**
> We thank the reviewer for the observation. Although our method introduces pairwise $\Delta_{ij}$ (quadratic in batch size), we **never store all deltas**. Instead, a **batch-streaming strategy** keeps memory practical:
>
> - For each 128×128 text–video block, $\Delta_{ij}$ (≈32MB in float32) is computed on-the-fly by $\psi$, used to form the similarity matrix, and immediately discarded;
> - Only one delta tensor exists at a time, and only the **similarity block** is retained for later concatenation.
>
> This design keeps memory low and preserves **dual-tower efficiency**. Runtime and memory (4×RTX 4090) on MSR-VTT are:
>
> |**Metric**|**CLIP4Clip**|**GARE**|
> |-|-|-|
> |Training Time|1h 30min| 1h 34min|
> |Inference Time|7.6s| 6.9s|
> |Training Memory|4 × 12175MB|4 × 12561MB|
> |Inference Memory|4136MB | 4216MB|
> |Training FLOPs (batch-wise)|39,167.58 GFLOPs|39,287.55 GFLOPs|
> |Inference FLOPs (batch-wise)|11,868.70 GFLOPs|11,905.05 GFLOPs|
>
> **Takeaway:**
> - $\Delta_{ij}$ is **transient**, only the similarity matrix is stored, and overhead remains minimal.
> - Training memory increases by **~3%**, and inference memory increases by ~1%, which is modest.
> - $\Delta$ is computed via **pair-wise paralleled** cross attention (details in ztES).
>
> *(Due to space constraints, the detailed logic of the inference stage can be found in my response to Reviewer ztES. I apologize again for this.)*
>
> -----
>
> ### **Hard Negative Comparison**
>
>  We compare GARE with two recent methods related to hard negative handling:
>
> **DMAE** (ACM MM 2023, R@1: 46.9, +1.6% over base, our reproduction: 47.1):
>  DMAE improves fine-grained alignment by mining hard positives (e.g., queries tied to specific frames), which implicitly enhances the model’s ability to push away hard negatives. This is conceptually similar to our *IB loss*, which compresses $\Delta$ via a variational bottleneck to retain critical alignment signals.
>
> **NeighborRetr** (CVPR 2025, R@1: 49.5, +2.3% over base, our reproduction: 49.2):
>  Uses a memory bank to compute k-neighbor co-occurrence and identify “good hubs.” This promotes local consistency and reduces over-penalization of hard negatives. While not explicitly framed as hard negative mining, selecting top-k co-occurring samples achieves similar effects.
>
> **GARE** (R@1: 49.1, +2.5% over base):
>  GARE does **not require explicit hard negative mining**. Each $\Delta_{ij}$ absorbs loss gradients locally from the corresponding pair, reducing reliance on $t_i$ when encountering hard negatives. This softens noisy updates and improves generalization.
>
> **Efficiency Comparison**:
>
> - GARE uses only one cross-attention layer; NeighborRetr includes 8 MLPs and multiple transformer/conv layers.
> - NeighborRetr requires 10240-sample memory banks per modality and ~4.5h training time; GARE achieves similar performance with **1h34min** training and minimal memory.
>
> Empirically, we observe that $t_i$ in GARE becomes semantically more stable: for semantically similar $v_j$, the similarities $s(t_i, v_i)$ and $s(t_i, v_j)$ are smoother than in the baseline. This supports more accurate delta-based alignment and aligns with Fig. 2, where $t_i$ functions as a stable prototype. Ultimately, the enhanced semantic stability of $t_i$ allows $\Delta$ to carry out more precise fine-grained alignment across similar samples. (Due to rebuttal guidelines, we regret that we cannot include figures here.)
>
> ## **Q3: Role of IB loss**
>
> Our method follows the **variational information bottleneck (VIB)** principle.
>  The standard VIB objective is
>
> $\mathcal{L}\_{\text{VIB}} = -\mathbb{E}_{z\sim p(z|x)}[\log q(m|z)] + \beta \cdot \mathrm{KL}\big(p(z|x) || r(z)\big),$
>
> where $x$ is the input, $z$ is the bottleneck variable, and $m$ is the match label.
>  In our cross-modal retrieval setting, we set $x = (t_i, v_j)$ and $z = \Delta_{ij}$.
>  The first term naturally corresponds to the **InfoNCE loss** on positive/negative text–video pairs, and the second term corresponds to our **IB loss (KL divergence)** on $\Delta_{ij}$.
>
> In principle, the compression term should be **pair-wise**:
>
> $\mathrm{KL}\big(p(\Delta_{ij}\mid t_i,v_j) || r(\Delta)\big),$
>
> which would require modeling each $\Delta_{ij}$ as a distribution and sampling $k$ times, leading to $(k, B_t, B_v, \text{dim})$ complexity.
>  To make this tractable, we adopt a **video-conditioned approximation**:
>
> $p(\Delta_{ij}\mid t_i,v_j) \approx p(\Delta_{ij}\mid v_j),$
>
> and compute the KL term by taking the mean and standard deviation along the text dimension:
>
> $\mu_j = \mathrm{mean}\_i(\Delta_{ij}), \quad \sigma_j = \mathrm{std}\_i(\Delta_{ij}),$
>
> approximating $p(\Delta_{ij}\mid v_j) \approx \mathcal{N}(\mu_j, \sigma_j^2)$.
>  This is consistent with the **many-to-many nature** of video–text datasets (e.g., MSRVTT has ~20 captions per video) and still satisfies the Monte Carlo property by treating text samples as stochastic draws. Empirically, video-conditioning outperforms text-conditioning because videos provide more stable, modality-level common information that forms an effective bottleneck.
>
> Our **overall training objective** is therefore:
>
> $\mathcal{L}\_{\text{total}} = \underbrace{\mathcal{L}\_{\text{Info}} + \beta \cdot \mathcal{L}\_{\text{IB}}}\_{\text{VIB objective}} + \lambda_{\varepsilon} \cdot \mathcal{L}\_{\varepsilon} + \lambda_{\text{dir}} \cdot \mathcal{L}\_{\text{dir}},$
>
> where we set $\beta = 0.07$, and $\lambda_{\varepsilon} = \lambda_{\text{dir}} = 0.01$.
>  The small weights for norm/direction regularization are due to the naturally large vector magnitudes in a 512-D embedding space, where a small coefficient is sufficient for stable adjustment.
>
> Finally, this also explains **why norm and direction regularization only take effect when combined with IB loss**.
>  Without the IB loss, $\Delta_{ij}$ has excessive freedom under InfoNCE and can trivially vary in magnitude or orientation, making these penalties ineffective.  The IB loss compresses $\Delta_{ij}$ into a **low-entropy, structured bottleneck**, which stabilizes its distribution; under this constraint, the norm and direction terms provide meaningful geometric guidance to improve alignment and uniformity.

---

> ### Author Response · Authors · 2025-08-05
>
> Dear Reviewer,
>
> I hope this message finds you well. As the discussion period is nearing its end with **less than three days remaining**, I wanted to ensure we have addressed all your concerns satisfactorily. If there are any additional points or feedback you'd like us to consider, please let us know. Your insights are invaluable to us, and were eager to address any remaining issues to improve our work.
>
> Thank you for your time and effort in reviewing our paper.

---

> ### Author Response · Authors · 2025-08-08
>
> Dear Reviewer,
>
> I hope this brief follow-up is not an inconvenience, and I apologize for reaching out once more. As the discussion period is nearing its close (approximately one day remaining), I wished to gently check whether there are any additional comments or clarifications I might address. I sincerely value your perspective. If time allows, any guidance you can share would be greatly appreciated and will inform our final revisions.
>
> I also understand that reviewer participation during the discussion phase is expected and encouraged under the NeurIPS process, as it helps ensure a balanced and thorough assessment. I would be grateful for any feedback at your convenience and look forward to hearing from you.
>
> Thank you again for your time and thoughtful consideration.

---

### Official Review · Reviewer_FbVo · 2025-07-02

**Clarity:** 3
**Significance:** 3
**Originality:** 3
**Rating:** 4
**Confidence:** 3

**Summary:**

The paper proposes GARE, a new gap-aware retrieval framework that introduces learnable pair-specific increments (Δij) to alleviate optimization tension in text-video contrastive learning. Through first-order Taylor approximation of InfoNCE loss, the work derives Δij as an optimal adjustment direction that resolves gradient conflicts caused by modality gaps and false negatives. The framework employs a lightweight neural module to predict Δij based on semantic gaps between pairs, enabling structure-aware corrections while maintaining training stability. Three key regularizations (trust-region constraint, directional diversity, and information bottleneck) are introduced to enhance the interpretability and geometric properties of the learned increments. Extensive experiments across four benchmarks demonstrate consistent improvements in alignment accuracy and robustness, with qualitative analysis showing Δij effectively redistributes gradient tension while preserving semantic structure.

**Questions:**

See Weaknesses section. I look forward to the author's response to my concerns during the rebuttal stage.

**Ethical Concerns:**

["NO or VERY MINOR ethics concerns only"]

**Final Justification:**

The rebuttal effectively addressed my concerns regarding efficiency and generalizability:

* Due to the use of a batch-wise streaming strategy, GARE is comparable in efficiency to CLIP4Clip.
* GARE also demonstrated performance improvements over the baseline in the Image-Text Retrieval task.

My primary concern remains with the method's adaptivity. The author provided experimental results for bidirectional delta generation along with a detailed analysis, concluding that the two-path module will weaken its alignment behavior, and it is indeed necessary to decide in which modality to inject Δij based on the characteristics of the dataset.

In summary, this paper proposes an innovative method to address the inherent optimization tension in contrastive learning, supported by thorough theoretical analysis and extensive experiments that validate the method's effectiveness. From my perspective, it is a work worthy of acceptance. Since my initial score was positive, I have decided not to change the score.

**Limitations:**

yes

**Paper Formatting Concerns:**

Writing standards and typos:
   * Equations 2 and 3 exhibit obvious subscript and superscript errors. For example, "$i$" should be a subscript, and "$B$" should not be a superscript of "$i=1$" but rather should be written as $\sum_{j=1}^{B}$.
   * Line 268: "MST-VTT" should be corrected to "MSR-VTT."
   * The authors are advised to carefully review the article to correct these typos and formula formatting issues.

**Quality:**

3

**Strengths And Weaknesses:**

**[Strengths]**

**+ Novel solution.** The paper introduces a novel approach to address the inherent optimization tension in contrastive learning by proposing pair-specific increments (Δij). This mechanism dynamically redistributes gradient conflicts caused by modality gaps and false negatives, offering a more stable and interpretable optimization landscape.

**+ Theoretical support.** The work derives the ideal form of Δij through a first-order Taylor expansion under trust-region constraints, providing a solid theoretical foundation.

**+ Extensive experiments.** The paper demonstrates consistent improvements across four diverse benchmarks (MSR-VTT, DiDeMo, ActivityNet, MSVD), outperforming recent state-of-the-art methods. The ablation studies (e.g., modality choice, regularization strength) provide actionable insights for adapting the framework to different data characteristics.

**+ Qualitative analysis.** The visualization reveals that Δij learns geometrically structured corrections—pushing embeddings outward while maintaining angular separation. This aligns with the uniformity principle of contrastive learning.

**[Weaknesses]**

**- Lack of efficiency analysis.** The application of learnable per-pair increments inevitably introduces additional training and inference overhead. However, the article only provides qualitative and quantitative analysis of this method on the Text-Video Retrieval task. Does the performance improvement solely come from adding more learnable parameters?

**- The method lacks adaptability.** According to the left side of Table 3 in the ablation experiments section, the authors extensively explored the Context Modality Choice and concluded that the decision on which modality to inject Δij into should be based on the characteristics of the dataset. Moreover, the reported results for each dataset in the article require selecting the best-performing setting through trial and error. This raises concerns about the method's poor generalizability and universality, as it necessitates dataset-specific configurations. Can this process be made adaptive?

**- More tasks are needed to verify generalizability.** Due to the presence of annotation noise, the issue of optimization tension is not limited to contrastive learning in text-video retrieval tasks but also exists in various cross-modal contrastive learning tasks. It is recommended that the authors conduct additional experiments on tasks such as text-image retrieval to verify whether the proposed method possesses sufficiently strong generalizability.

Writing standards and typos:
   * Equations 2 and 3 exhibit obvious subscript and superscript errors. For example, "$i$" should be a subscript, and "$B$" should not be a superscript of "$i=1$" but rather should be written as $\sum_{j=1}^{B}$.
   * Line 268: "MST-VTT" should be corrected to "MSR-VTT."
   * The authors are advised to carefully review the article to correct these typos and formula formatting issues.

---

> ### Author Rebuttal · Authors · 2025-07-29
>
> Thank you for raising the questions and carefully pointing out the incorrect writings! We will correct them and check the spelling again. Below is my response to the questions.
>
> ---
>
> ## **Q1: Impact of Efficiency and Parameter Quantity**
> We thank the reviewer for the insightful observation and concerns about the efficiency of the method. While our method introduces pairwise $\Delta_{ij}$ with quadratic complexity, we **do not store all deltas**. Instead, we use a **batch-wise streaming strategy** that keeps memory usage practical.
>
> At inference time, we precompute all CLIP encoder outputs and evaluate cosine similarities between text–video batches (e.g., 128×128 pairs) using the lightweight delta module $\psi$. For each batch pair:
>
> - $\Delta_{ij}$ (shape: 128×128×512, ≈32MB in float32) is computed on-the-fly;
> - It is immediately discarded after computing the 128×128 similarity matrix;
> - Only the similarity block is retained and later concatenated into the full similarity matrix.
>
> Thus, **only one delta tensor is active at any time, and is Pair-Wise parallelization**.
>
> Below is a runtime and memory comparison (4×RTX 4090 GPUs) on MSR-VTT:
>
> |**Metric**|**CLIP4Clip**|**GARE**|
> |-|-|-|
> |Training Time|1h 30min|1h 34min|
> |Inference Time|7.6s| 6.9s|
> |Training Memory|4 × 12175MB|4 × 12561MB|
> |Inference Memory|4136MB|4216MB|
> |Training FLOPs (batch-wise)|39,167.58 GFLOPs|39,287.55 GFLOPs|
> |Inference FLOPs (batch-wise)|11,868.70 GFLOPs|11,905.05 GFLOPs|
>
> **Key takeaway**:
>
> - $\Delta_{ij}$ is a **transient variable**, not stored;
> - Only the similarity matrix is retained;
> - $\Delta$ is computed via **pair-wise paralleled** cross attention (details in ztES).
>
> *(Due to space constraints, more details about pair-wise parallelization computing can be found in my response to Reviewer ztES. I apologize again for this.)*
>
> ---
>
> ### **Impact of parameter quantity**
>
> Thank you for raising this concern. To clarify that the improvements in GARE are not merely due to the increase in parameters, we conducted several experiments comparing GARE with the baseline (**while the performance of GARE: R@1=49.1,R@5=74.7,R@10=83.6**):
>
> 1. **Baseline comparison with additional layers**: We increased the **video temporary Transformer** from 4 to 5 layers in the baseline, aligning its parameters with GARE. The performance was **R@1 = 47.3**, **R@5 = 73.2**, **R@10 = 82.5**, showing that the increase in parameters led to only a modest improvement, indicating GARE’s better performance with the same parameter budget.
> 2. **Parameter alignment with baseline**: Reducing the **video temporary Transformer** layers in GARE to 3, we aligned the parameter count with the baseline. The results were **R@1 = 48.1**, **R@5 = 73.8**, **R@10 = 83.6**, confirming that GARE has a structural advantage.
> 3. **Interaction modification**: We modified GARE’s interaction by setting the **query** for $\psi$ to be $t_i$ and the **key-value** to be frames from $v_j$, outputting $t_i'$. To match the scale of the delta in GARE, we computed $\Delta = v_j - t_i'$ (or $t_i' - v_j$), effectively removing the relative structure information between $t_i$ and $v_j$. This modification resulted in **R@1 = 47.6**, **R@5 = 73.1**, **R@10 = 82.0**, **MdR = 2.0**, and **MnR = 12.6**, showing that while GARE uses a pair-wise interaction, its **interaction logic** is superior. The results demonstrate that **$\psi$'s awareness of the relative structure between $v_j$ and $t_i$** is indeed crucial for improving performance.
>
> In conclusion, GARE’s performance gains come from **structural advantages** and **improved interaction logic**, not just the addition of parameters (only **1.5M**, far less than CLIP’s **151M**).
>
> # **Q2: Adaptivity of the Method**
>
> Thank you for raising the question about adaptivity. To explore this, we intuitively used two $\psi$ modules for bidirectional delta generation, resulting in $(t_i + \Delta^{t}\_{ij}, v_j + \Delta^{v}\_{ij})$. This led to performance degradation, with R@1 dropping from 42.6 to 41.2 on ActivityNet Captions and from 49.1 to 48.4 on MSR-VTT, though still outperforming the baseline, indicating the delta-based approach maintains adaptivity. Additionally, we observed performance improvement in an image-text retrieval task on MSCOCO, further supporting GARE’s adaptivity across different domains (details in the final answer).
>
> We hypothesize that this decline results from **progressive weakening of $\Delta^{t}_{ii}$'s alignment role** due to **semantic ambiguity introduced by $\Delta^{v}\_{ii}$**. As training progresses, $v_i$ evolves into a **prototype-style anchor**, similar to Fig. 2(b). When $v_i$ is used as the context in $\psi$ to compute $\Delta^t_{ii}$, its **over-generalized nature** hinders accurate computation of $\Delta^t_{ii}$, reducing alignment effectiveness and compromising training.
>
> To validate that $\Delta_{ii}$ performs the alignment task, we **compare the learned delta update magnitude with the theoretically derived update magnitude**, measuring the error to assess whether $\Delta_{ii}$'s behavior aligns with the theoretical expectation. By expanding $\mathcal{L}\_i$ w.r.t. $\Delta_{ij}^{(t)}$, we obtain the **analytic update step size** $\alpha_{ij}^{(t)}$, ensuring $\Delta_{ij}^{(t+1)}$ stays within a trust region:
>
> $\left\| \Delta_{ij}^{(t)} - \alpha_{ij}^{(t)} \hat{g}\_{ij}^{(t)} \right\|^2 \le \varepsilon^2, \quad  \hat{g}\_{ij}^{(t)} = \frac{\nabla_{\Delta_{ij}} \mathcal{L}\_i^{(t)}}{\|\nabla_{\Delta_{ij}} \mathcal{L}\_i^{(t)}\|}, \varepsilon = \|\Delta_{ij}^{(t+1)}\|.$
>
> Simplifying for the boundary case gives the **analytic step size**:
>
> $\alpha_{ij}^{(t)} = \Delta_{ij}^{(t)\top} \hat{g}\_{ij}^{(t)} + \sqrt{ \left( \Delta_{ij}^{(t)\top} \hat{g}\_{ij}^{(t)} \right)^2 - \|\Delta_{ij}^{(t)}\|^2 + \varepsilon^2 }.$
>
> In contrast, the actual update induced by $\psi$ after one optimizer step is:
>
> $\Delta_{ij}^{(t+1)} = \Delta_{ij}^{(t)} - \eta_{ij}^{(t)} \cdot \frac{\nabla_{\Theta^{(t)}} \mathcal{L}\_i^{(t)}}{\|\nabla_{\Theta^{(t)}} \mathcal{L}\_i^{(t)}\|},$
>
> where $\eta_{ij}^{(t)}$ is the implicit learned step size. To evaluate their consistency, we compute the error between the two step sizes:
>
> $|\eta_{ij}^{(t)} - \alpha_{ij}^{(t)}|.$
>
> ----
>
> - Empirical results show that for **positive pairs** ($i = j$), the error remains within a **moderate range** ([1.0, 4.5] in 512-D space) and **decreases steadily** as training progresses, indicating that $\psi$ produces updates **consistent with the theoretical rule**. This validates that $\Delta_{ii}$ plays a **precise alignment role**.
>
> - For **negative pairs** ($i \neq j$), the error is larger and does not converge, as $\Delta_{ij}$ receives only **push gradients from $v_j$** without alignment supervision. This optimization occurs in a **$(D-1)$-dimensional space**, leading to more variable updates compared to the **1-dimensional alignment** of $\Delta_{ii}$. Consequently, $\Delta_{ij}$ primarily enhances **uniformity** by increasing scale and dispersing representations.
>
> - **Comparison with Dual $\psi$ Modules**:
>  When using dual $\psi$ modules (i.e., $(t_i + \Delta^t_{ij}, v_j + \Delta^v_{ij})$), both $\Delta^t_{ii}$ and $\Delta^v_{ii}$ show **slower convergence** to the theoretical update step size $\eta^{(t)}$. Surprisingly, after some initial convergence, the update magnitude error for $\Delta^t_{ii}$ and $\Delta^v_{ii}$ **starts to diverge**, which suggests that their **alignment capabilities are weakened**, preventing accurate update directions from being learned.
>
> ---
>
> This validates our analysis: when $\Delta_{ii}$ performs the alignment function, using the two-path $\psi$ module will weaken its alignment behavior. In future work, we will focus on alleviating the optimization tension while seeking a more adaptive dual-path method.
>
> *(We will include the error plots in the appendix and public codebase if the paper is accepted, as current rebuttal guidelines do not permit figures.)*
>
> # **Q3: GARE' performance on Image-Text Retrieval**
> Thank you for your suggestion to evaluate GARE’s generalization in image-text retrieval. We used CLIP as the baseline and conducted ITM tasks on MS-COCO 1K and 5K, comparing it with methods like VSE∞, PCME++, and PAU. Due to time constraints, we did not fine-tune GARE on MS-COCO, resulting in limited improvement on the ITM task. Currently, we use CLIP’s image patch tokens as the context input for the module, but these often introduce noise and weaker semantic information compared to the image’s `[CLS]` embedding. In the future, we plan to aggregate the patches to extract cleaner features, enabling better validation of GARE’s effectiveness on the ITM task. Below are the results of GARE on MS-COCO with ViT-B/32 as the backbone:
>
> |Method||1k|i2t||1k|t2i||5k|i2t||5k|t2i|
> |-|-|-|-|-|-|-|-|-|-|-|-|-|
> ||R@1|R@5|R@10|R@1|R@5|R@10|R@1|R@5|R@10|R@1|R@5|R@10|
> |VSE∞|**82.0**|**97.2**|98.9|_69.0_|_92.6_|_96.8_|62.3|**87.1**|**93.3**|**48.2**|**76.7**|**85.5**|
> |PCME++|_81.6_|**97.2**|_99.0_|**69.2**|**92.8**|**97.1**|62.1|_86.8_|**93.3**|_48.1_|**76.7**|**85.5**|
> |PAU|80.4|96.2|98.5|67.7|91.8|96.6|_63.6_|85.2|92.2|46.8|74.4|*83.7*|
> |**Baseline**|80.1|95.7|98.2|67.1|91.4|96.6|62.9|84.9|91.6|46.5|73.8|82.9|
> |**ours**|80.7|_96.8_|**99.1**|68.2|91.9|96.6|**63.9**|85.5|*92.4*|47.2|*74.8*|83.5|
>
> At the same time, we also performed **noise correspondences** training on MS-COCO, which demonstrates significantly better robustness compared to the baseline:
>
> |Noise ratio|Method||1k|i2t||1k|t2i||5k|i2t||5k| t2i|
> |-|-|-|-|-|-|-|-|-|-|-|-|-|-|
> |||R@1|R@5|R@10|R@1|R@5|R@10|R@1|R@5|R@10|R@1|R@5|R@10|
> |20%|Baseline|76.0|94.3|97.5|63.4|89.0|94.8|55.3|79.1|86.9|41.0|68.8 |79.3|
> ||ours|**79.4**|**96.0**|**98.8**|**65.2**|**90.7**|**96.4**|**58.9**|**83.7**|**91.2**|**43.7**|**70.7**|**80.8**|
> |50%|Baseline|73.9|93.0|97.2|60.1|87.3|94.0|54.1|78.5|86.6|39.7|67.2|77.5|
> ||ours|**77.2**|**93.6**|**97.2**|**62.5**|**89.1**|**94.2**|**56.1**|**80.8**|**87.9**|**41.2**|**68.6**|**79.2**|

---

> > ### Comment · Reviewer_FbVo · 2025-08-07
> >
> > Thank you very much for detailed reply. The rebuttal effectively addressed my concerns regarding efficiency and generalizability:
> >
> > * Due to the use of a batch-wise streaming strategy, GARE is comparable in efficiency to CLIP4Clip.
> > * GARE also demonstrated performance improvements over the baseline in the Image-Text Retrieval task.
> >
> > My primary concern remains with the method's adaptivity. The author provided experimental results for bidirectional delta generation along with a detailed analysis, concluding that the two-path module will weaken its alignment behavior, and it is indeed necessary to decide in which modality to inject Δij based on the characteristics of the dataset.
> >
> > In summary, this paper proposes an innovative method to address the inherent optimization tension in contrastive learning, supported by thorough theoretical analysis and extensive experiments that validate the method's effectiveness. From my perspective, it is a work worthy of acceptance. Since my initial score was positive, I have decided not to change the score.

---

> > > ### Author Response · Authors · 2025-08-07
> > >
> > > Thank you for your thoughtful comments and for highlighting the importance of evaluating generalization and adaptivity in image-text retrieval tasks.
> > >
> > > We truly appreciate your insightful suggestions, which helped broaden the scope of our evaluation and strengthen the practical significance of our method. We have taken your advice seriously and will include corresponding experiments and discussions in the revised manuscript and appendix.
> > >
> > > Thank you again for your valuable feedback and support throughout the review process.

---

> ### Author Response · Authors · 2025-08-05
>
> Dear Reviewer,
>
> I hope this message finds you well. As the discussion period is nearing its end with **less than three days remaining**, I wanted to ensure we have addressed all your concerns satisfactorily. If there are any additional points or feedback you'd like us to consider, please let us know. Your insights are invaluable to us, and were eager to address any remaining issues to improve our work.
>
> Thank you for your time and effort in reviewing our paper.

---

### Official Review · Reviewer_ztES · 2025-07-03

**Clarity:** 3
**Significance:** 2
**Originality:** 3
**Rating:** 3
**Confidence:** 3

**Summary:**

The paper proposes GARE, a framework addressing optimization challenges in text-video retrieval by introducing learnable pair-specific increments $\Delta_{i,j}$ to mitigate gradient tension from modality gaps and false negatives. The paper presents a theoretically grounded approach to addressing gradient tension in text-video retrieval, with strong experimental support for its effectiveness. However, its departure from the dual-tower paradigm and potential inefficiencies raise critical questions about its practical utility.

**Questions:**

please see weaknesses above

**Ethical Concerns:**

["NO or VERY MINOR ethics concerns only"]

**Limitations:**

please see weaknesses above

**Quality:**

4

**Strengths And Weaknesses:**

## Strengths
1. **Valid Core Motivation**: The work effectively targets a critical and well-recognized issue in cross-modal learning: the domain (modality) gap between text and video representations. By framing this gap as a source of optimization tension under contrastive learning, the paper identifies a meaningful problem with clear practical relevance, laying a solid foundation for the proposed solution.
2. **Rigorous Experimental Design**: The experiments are comprehensive and well-structured. The evaluation spans four standard benchmarks (MSR-VTT, DiDeMo, ActivityNet Captions, MSVD), with consistent comparisons to state-of-the-art methods. Ablation studies systematically validate the contributions of key components ($\Delta_{i,j}$, regularization terms, modality-specific application), ensuring the observed improvements are attributable to the proposed mechanisms rather than confounding factors.
3. **Sound Theoretical and Design Contributions**: The derivation of $\Delta_{i,j}$ via multivariate Taylor expansion of the InfoNCE loss under a trust-region constraint is mathematically rigorous, providing a clear rationale for pair-specific increments. The analysis of $\Delta_{i,j}$’s geometric properties (e.g., angular distributions) and the design of complementary regularizers (norm constraints, directional diversity, information bottleneck) demonstrate careful consideration of how to stabilize and interpret the learned increments. This integration of theory and design strengthens the framework’s validity.

## Weaknesses
1. **Violation of Dual-Tower Architecture Principles**: A central drawback is that the introduction of pair-specific $\Delta_{i,j}$ disrupts the independence of text and video towers in cross-modal retrieval. Traditional dual-tower models enable efficient retrieval via precomputed embeddings and dot-product similarity, but GARE’s $\Delta_{i,j}$ introduces dynamic, pair-dependent interactions $t_i + \Delta_{i,j}$ that cannot be precomputed. This undermines the core advantage of dual-tower designs—scalable, offline retrieval—and limits practical deployment in real-world systems where low latency is critical. The paper does not address this trade-off or propose mitigations (e.g., approximation strategies for faster inference).
2. **Efficiency Concerns**: The training and inference efficiency of GARE is inadequately discussed. The learnable $\Delta_{i,j}$ is pair-specific, meaning its computational cost scales with the number of text-video pairs in a batch, which could become prohibitive for large datasets or high-throughput applications. If $\Delta_{i,j}$ requires real-time computation for each query-target pair during inference, this would drastically slow down retrieval compared to static embedding-based methods. Without details on latency or scalability, the practical feasibility of GARE remains uncertain. Additionally, the paper does not clarify how $\Delta_{i,j}$ is computed for unseen test pairs.

---

> ### Author Rebuttal · Authors · 2025-07-26
>
> We sincerely thank the reviewer for raising two critical concerns:
> (1) whether the introduction of pair-specific deltas violates the principles of dual-tower architectures;
> (2) whether the proposed `ψ` module could impact training and inference efficiency.
>
> We address both points in detail below.
>
> ---
>
> # **Q1: Dual-Tower Compatibility**
>
> GARE augments a standard **dual-tower CLIP** architecture with a **lightweight cross-modal adjustment module** `ψ`.
>
> * `ψ` is implemented as a **single-layer cross-attention Transformer** with FFN hidden dimension not expanded, adding only **1.58M parameters** compared to the **354M parameters** in the CLIP encoder. At the same time, during the inference phase, the GFLOPs per batch introduced by `ψ` is **36.35 GFLOPs**, much smaller than the baseline encoder’s **11,868.70 GFLOPs** per batch.
> * It operates **after embeddings are precomputed** by the dual towers, applying pair-specific deltas to adjust text embeddings before computing cosine similarity.
>
> Importantly, this design **does not compromise dual-tower precomputability**:
>
> 1. **All text and video embeddings** are computed and cached offline by the CLIP encoder (4-layer video temporary Transformer equipped after CLIP vision encoder).
> 2. `ψ` only introduces a **transient delta computation** for each text–video batch pair.
> 3. **No delta embedding is stored**, and only the resulting cosine similarity block is retained.
>
> Thus, GARE **preserves the key advantages of dual-tower architectures**—offline caching and scalable batch-parallel retrieval—while enabling **fine-grained pairwise alignment** through a minimal adjustment step.
>
> ---
>
> # **Q2: Training and Inference Efficiency**
>
> Thank you for raising concerns about computational efficiency. We will address them by comparing FLOPs, memory usage, how efficient batch-pair level similarity calculation is implemented during inference, and how the $\psi$ module ensures negligible latency by computing (text, video) pair-wise parallelizable cross-attention.
>
> We provide a detailed comparison of GARE and CLIP4Clip on **4×RTX 4090 GPUs (batch size 128)** and MSR-VTT:
>
> | **Metric**                   | **CLIP4Clip**    | **GARE**                 |
> | ---------------------------- | ---------------- | ------------------------ |
> | Training Time                | 1h 30min         | 1h 34min                 |
> | Inference Time               | 7.6s             | 6.9s                     |
> | Training Memory (reserved)   | 4 × 12175MB      | 4 × 12561MB              |
> | Inference Memory (reserved)  | 4136MB           | 4216MB                   |
> | Training FLOPs (batch-wise)  | 39,167.58 GFLOPs | 39,287.55 GFLOPs (+0.3%) |
> | Inference FLOPs (batch-wise) | 11,868.70 GFLOPs | 11,905.05 GFLOPs (+0.3%) |
>
> ---
>
> ### **Module-wise GFLOPs Breakdown**
>
> Here is a summary of the **GFLOPs per module** during forward propagation:
>
> | Module                      | GFLOPs   | Percentage | Main Calculation Source            |
> |-----------------------------|----------|------------|------------------------------------|
> | CLIP Visual Encoder          | 11,673.60 | 98.05%     | ViT Image Processing               |
> | Video Temporal Transformer   | 38.81    | 0.33%      | FFN (66.4%) + Self-Attention (33.3%) |
> | `psi` (Cross-Attn Transformer)  | 36.35    | 0.31%      | Linear Projection (73.6%)          |
> | CLIP Text Encoder            | 156.29   | 1.31%      | Text Sequence Processing           |
> | **Total**                    | **11,905.05** | **100%**  | -                                  |
>
> ---
>
> **Key observations:**
>
> * Training memory increases by **\~3%**, and inference memory increases by **\~1%**, which is modest.
> * FLOPs increase in inference is **\~0.3% per batch pair** and remains dominated by the encoder.
> * End-to-end inference is **slightly faster** (6.9s vs 7.6s) due to efficient batch-wise similarity computation. **Delta computation is identical for training and inference and remains transient (e.g., 128×128×512, \~32MB) during the forward pass.** Only the cosine similarity block is retained for further computation.
>
> Crucially, **deltas are computed on-the-fly and immediately discarded**, with a typical tensor (128×128×512, float32) occupying only **≈32MB** transiently. This **streaming computation** ensures that memory usage remains minimal even for large-scale datasets.
>
> ---
>
> ### **Inference Procedure (Pseudocode)**
>
> The inference pipeline preserves dual-tower efficiency via **batch-parallel streaming computation**:
>
> ```python
> # Precompute all text and video embeddings offline using CLIP encoders
>
> for each text batch T:
>     for each video batch V:
>         # 1. Compute pair-specific delta via ψ and adjust text embeddings
>         logits = model.get_similarity_logits(T, V)
>
>         # 2. Append cosine similarity block to the global similarity matrix
>         sim_matrix.append(logits)
>
>         # 3. Discard the delta tensor immediately (transient variable)
>
> # Concatenate all similarity blocks for final retrieval
> ```
>
> **Key points:**
>
> * Deltas are **never stored**, only used to produce logits;
> * Memory footprint remains low, and computation is **fully GPU-parallel**;
> * The design is **deployable for real-world retrieval** with minimal latency.
>
> ---
>
> ### **Cross-Attention Pair-Wise Parallelization**
>
> To further clarify how the cross-attention mechanism in \$\psi\$ operates efficiently, here is the **pseudocode** for the pair-wise parallelized cross-attention computation:
>
> ```python
> def CrossAttentionLayer(query, key, value):
>     """
>     Compute Q, K, V of Cross-Attention (omitting multi-head).
>
>     Args:
>         query: ti - vj with shape (a, b, dim)
>         key & value: video context sequence with shape (b, f, dim), where f is the number of frames
>
>     Returns:
>         Attention output
>     """
>
>     a, b, dim = query.shape
>     _, f, _ = key.shape
>
>     query = query.permute(1, 2, 0)  # -> (b, dim, a)
>
>     q_proj = Q_proj(query)
>     k_proj = K_proj(key)
>     v_proj = V_proj(value)
>
>     # This operation can be pair-wise parallelized
>     attn_logits = k_proj @ q_proj  # (b, f, dim) x (b, dim, a) -> (b, f, a)
>     attn_scores = softmax(attn_logits / sqrt(d), dim=1).permute(0, 2, 1)  # -> (b, a, f)
>
>     attn_out = attn_scores @ v_proj  # (b, a, f) x (b, f, dim) -> (b, a, dim)
>     attn_out = Out_proj(attn_out).permute(1, 0, 2)  # -> (a, b, dim)
>
>     return attn_out
> ```
>
> **Key points:**
>
> * This computation is **pair-wise parallelizable**, meaning it processes text-video pairs within a batch efficiently, leveraging parallelization across the batch dimension.
> * This design ensures that the cross-attention operation remains **scalable and efficient**, preserving the advantages of dual-tower retrieval while introducing fine-grained alignment.
>
> ---
>
> # **Conclusion**
>
> By embedding `ψ` in a **streaming, transient, and batch-parallel** workflow, GARE:
>
> 1. **Preserves dual-tower scalability and offline precomputability**;
> 2. **Adds negligible memory and FLOPs overhead**;
> 3. **Maintains low-latency retrieval**, compatible with large-scale deployments;
> 4. **Provides expressive pair-specific alignment** that significantly improves retrieval performance.
>
> We appreciate the reviewer’s feedback, which helped us clarify GARE’s **practical efficiency and deployability**.

---

> > ### Comment · Reviewer_ztES · 2025-08-06
> >
> > I appreciate the authors' detailed response and thorough analysis of the training and inference time costs, including specific metrics like FLOPs, runtime, and memory usage. These breakdowns help contextualize the work’s practical efficiency, and I acknowledge the effort put into addressing these technical details. However, my core concern—namely, that this work violates the fundamental principles of Dual-Tower Architectures—remains unaddressed, and my broader efficiency worries are deeply intertwined with this issue.
> >
> > To clarify, violating Dual-Tower principles does not inherently lead to slower performance when measured against a naive $O(NM)$ matching baseline (where $N$ represents the number of queries and $M$ the number of documents). In fact, in some cases, adding interactive modules might even yield marginal speedups in controlled scenarios. The critical issue, though, lies in how Dual-Tower Architectures are designed to enable scalable, real-world retrieval systems. At their core, these architectures rely on the separation of query and document processing to facilitate approximate nearest neighbor (ANN) search—a cornerstone of efficient large-scale retrieval. ANN systems thrive on **static, precomputable *distance* metrics**: for instance, linear products that translate to cosine similarity or Euclidean distance. These metrics are optimized to reduce retrieval complexity from $O(NM)$ to $O(N log M)$, making them feasible for datasets with millions or billions of documents.
> >
> > A key reason these metrics work is their adherence to mathematical properties that underpin ANN’s efficiency: nonnegativity, identity, symmetry, and, crucially, the triangle inequality. These properties allow ANN algorithms to prune the search space effectively, avoiding exhaustive comparisons.
> >
> > By introducing a cross-modal adjustment module, however, this work disrupts this foundational framework. First, the module breaks the **triangle inequality**, a property that ANN systems depend on to approximate nearest neighbors efficiently. Second, it transforms the distance metric into a query-dependent measure—one that shifts based on the specific query being processed. This dynamism undermines the ability to precompute document embeddings or distance scores, which is essential for scaling retrieval to large document collections. Instead of leveraging precomputed indices, the system would need to recalculate interactions for each query, effectively reverting to a more computationally heavy paradigm that sacrifices the Dual-Tower Architecture’s primary advantage: scalable, low-latency retrieval.
> >
> > Thus, while the analysis of FLOPs and runtime provides useful insights, the deeper issue of violating Dual-Tower principles—and the consequent impact on scalability—remains a significant concern for the work’s practical applicability in large-scale retrieval scenarios.

---

> > > ### Author Response · Authors · 2025-08-09
> > >
> > > Dear Reviewer,
> > >
> > > I hope this brief follow-up is not an inconvenience, and I apologize for reaching out once more. As the discussion period is nearing its close (less than one day remaining), I wished to gently check whether there are any additional comments or clarifications I might address. I sincerely value your perspective. If time allows, any guidance you can share would be greatly appreciated and will inform our final revisions.
> > >
> > > I also understand that reviewer participation during the discussion phase is expected and encouraged under the NeurIPS process, as it helps ensure a balanced and thorough assessment. I would be grateful for any feedback at your convenience and look forward to hearing from you.
> > >
> > > Thank you again for your time and thoughtful consideration.

---

> ### Author Response · Authors · 2025-08-05
>
> Dear Reviewer,
>
> I hope this message finds you well. As the discussion period is nearing its end with **less than three days remaining**, I wanted to ensure we have addressed all your concerns satisfactorily. If there are any additional points or feedback you'd like us to consider, please let us know. Your insights are invaluable to us, and were eager to address any remaining issues to improve our work.
>
> Thank you for your time and effort in reviewing our paper.

---

> ### Author Response · Authors · 2025-08-06
>
> Thank you very much for your detailed comments and for highlighting the dual-tower efficiency concerns. We apologize for not providing a thorough discussion on ANN-based retrieval in the initial submission. Below, we present our refined analysis and solution.
>
> ---
>
> ### **1. Two-Stage ANN Retrieval with GARE**
>
> Although GARE introduces pair-dependent refinements $\Delta_{ij}$ via $\cos(t_i+\Delta_{ij}, v_j)$ during training, the dual-tower encoders still produce static embeddings $t_i$ and $v_j$, which are suitable for efficient ANN indexing. In fact, when we evaluate the dual-tower part (which is exactly CLIP4Clip [1], a classical dual-tower method in text video retrieval) on the MSR-VTT dataset after training with GARE, we obtain R\@1 = 44.2, R\@5 = 71.7, R\@10 = 81.2, MdR = 15.2, which is almost identical to the original CLIP4Clip results (44.5 / 71.4 / 81.6 / 15.3). This demonstrates that our dual-tower branch maintains retrieval stability, ensuring that it can still be reliably used for large-scale indexing via ANN. Based on this, we can apply GARE in large-scale scenarios using a two-stage retrieval pipeline: in the first stage, we build the ANN index with the dual-tower outputs of GARE and retrieve the Top-K candidates online; in the second stage, we apply the GARE's `ψ`-module to these Top-K candidates for refined re-ranking.
>
> * **Stage 1 (Candidate Retrieval)**
>
>   1. Normalize \$(t\_i, v\_j)\$ to construct the ANN index.
>   2. Store both normalized vectors and their L2 norms, because cosine similarity depends on direction, while GARE refinement also involves magnitude.
>   3. Perform ANN search using \$\cos(t\_i,v\_j)\$ to efficiently obtain Top-K candidates.
>
> * **Stage 2 (Δ-based Reranking)**
>
>   1. Retrieve the original (non-normalized) embeddings for the Top-K candidates.
>   2. Compute \$\Delta\_{ij}\$ with the ψ-module and refine similarities as \$\cos(t\_i+\Delta\_{ij}, v\_j)\$.
>   3. Rerank the Top-K to obtain the final retrieval results.
>
> This design maintains scalable ANN efficiency while leveraging Δ for fine-grained alignment.
>
> ---
>
> ### **2. Deviation Analysis of Δ**
>
> Because GARE modifies similarity to \$\cos(t\_i+\Delta\_{ij}, v\_j)\$, we analyze how Δ affects static dual-tower similarity \$\cos(t\_i,v\_j)\$.
>
> **Theoretical upper bound:**
> Let \$a=t\_i\$, \$b=v\_j\$, \$\delta=\Delta\_{ij}\$; on MSR-VTT-1k, \$|t\_i|\approx11.0\$, \$|\Delta|\approx3.31\$.
>
> $$
> \delta_{ij}
>   = \big|\cos(a+\delta,b)-\cos(a,b)\big|
>   \lesssim \frac{\|\delta\|}{\|a\|}
>   \approx 0.30
> $$
>
> This first-order approximation shows Δ induces ≤0.3 deviation in the worst case, and is smaller for most pairs.
>
> **Empirical deviation:** We evaluate \$|\cos(t\_i+\Delta\_{ij},v\_j)-\cos(t\_i,v\_j)|\$ across MSR-VTT-1k:
>
> 1. Mean Deviation (\$\mathbb{E}\[\delta\_{ij}]\$)
>
>    * Average deviation across all pairs: 0.084
>    * Meaning: Δ changes similarity by only 8.4% on average, indicating minimal perturbation.
>
> 2. 95th Percentile Deviation (\$P\_{95}\$)
>
>    * 95% of all deviations < 0.17
>    * Meaning: Δ refinement is locally smooth, and extreme deviations are rare.
>
> 3. Top-K Candidate Coverage (\$C\_K\$)
>
>    * Overlap between Top-10 by GARE and Top-10 by static dual-tower within Top-256 ANN candidates ≥ 99%.
>    * Meaning: A K=256 candidate pool almost always recovers the final GARE Top-10, confirming reliable ANN pre-filtering.
>
> These results show that static dual-tower embeddings remain highly reliable for ANN retrieval even with Δ refinement.
>
> ---
>
> ### **3. Stability of Static Embeddings**
>
> GARE also stabilizes the dual-tower outputs:
>
> * Hard negatives often produce noisy gradients on \$t\_i\$ in standard InfoNCE.
> * GARE absorbs part of this tension into Δ, reducing direct noise on \$t\_i\$.
> * Empirically, the variance of \$t\_i\$ decreases after GARE training, and static dual-tower retrieval remains competitive.
>
> This means dual-tower embeddings remain both stable and ANN-compatible.
>
> ---
>
> ### **4. Conclusion**
>
> * Static embeddings remain ANN-compatible with negligible retrieval loss.
> * Δ perturbations serve as lightweight, second-stage refinements with bounded deviation (≤0.3 theoretical, 0.084 empirical).
> * Two-stage retrieval with Top-K reranking preserves scalability while exploiting Δ for fine-grained alignment.
>
> These results confirm that GARE maintains the dual-tower paradigm for large-scale retrieval and that Δ is a practical, noise-resilient enhancement.
>
> ---
>
> **References**
>
> [1] Luo, Huaishao, et al. "Clip4clip: An empirical study of clip for end to end video clip retrieval and captioning." Neurocomputing 508 (2022): 293-304.

---

> ### Author Response · Authors · 2025-08-07
>
> Thank you again for your thoughtful response and continued attention to our work.
> We would also like to clarify that pair-wise interactions after dual-tower encoders constitute a relatively common framework in text-video retrieval. Below, we provide a detailed explanation.
>
>  In the domain of text-video retrieval, it is actually quite common for recent methods to go beyond the strict dual-tower architecture by introducing **cross-modal interaction modules after the independent encoders**. We list several representative examples below, all of which introduce pair-wise interaction after the dual-tower stage:
>
> - EMCL [5] (NeurIPS 2022): R@1 = 46.8, R@5 = 73.1, R@10 = 83.1
> - X-Pool [4] (CVPR 2022): R@1 = 46.9, R@5 = 72.8, R@10 = 82.2, MnR = 14.3
> - DiCoSA [2] (IJCAI 2023): R@1 = 47.5, R@5 = 74.7, R@10 = 83.8, MnR = 13.2
> - DiffusionRet [3] (ICCV 2023): R@1 = 49.0, R@5 = 75.2, R@10 = 82.7, MnR = 12.1
>
> All these methods are built upon the CLIP4Clip [1] dual-tower backbone and introduce cross-modal interactions after feature encoding. This reflects a broader trend in the field, where many methods have moved beyond strict dual-tower architectures to incorporate pair-wise alignment strategies that better capture co-occurrence patterns and fine-grained correspondences between text and video. Notably, these classical works have inspired a range of subsequent advancements, further refining such interaction modules and demonstrating their effectiveness in improving retrieval accuracy. This evolution highlights that, in text-video retrieval, enhancing cross-modal alignment remains a key research focus.
>
> In fact, within the academic context of text-video (or image-text) retrieval, the primary challenge lies not in scalable hierarchical retrieval with ANN techniques (e.g., $O(N_q \log N_d)$ complexity via clustering), but rather in improving semantic alignment across modalities.
>
> This motivates a large body of research focused on alignment modeling rather than large-scale indexing, such as:
>
> - Fine-grained alignment at the frame-word level, action-word level, prototype level, or multi-level fusion.
> - Cross-modal generation or fusion, e.g., DiffusionRet [3] reframes the retrieval objective from discriminative $p(v_j|t_i)$ to generative $p(t_i,v_j)$, which is also need pair-wise interaction after dual-tower stage.
> - Many-to-many alignment strategies that reduce the impact of false negatives and the entropy imbalance between text and video.
>
> These efforts primarily aim to address the difficulty of aligning semantically sparse text with visually redundant video content. While we acknowledge that adherence to dual-tower principles is foundational for large-scale retrieval efficiency, we note that in the current landscape of text-video retrieval, the primary research focus remains on improving cross-modal alignment and discriminative performance, rather than optimizing for large-scale ANN-friendly indexing.
>
> ---
>
> **References**
>
> [1] Luo, Huaishao, et al. "Clip4clip: An empirical study of clip for end to end video clip retrieval and captioning." Neurocomputing 508 (2022): 293-304.
>
> [2] Jin, Peng, et al. "Text-video retrieval with disentangled conceptualization and set-to-set alignment." arXiv preprint arXiv:2305.12218 (2023).
>
> [3] Jin, Peng, et al. "Diffusionret: Generative text-video retrieval with diffusion model." Proceedings of the IEEE/CVF international conference on computer vision. 2023.
>
> [4] Gorti, Satya Krishna, et al. "X-pool: Cross-modal language-video attention for text-video retrieval." Proceedings of the IEEE/CVF conference on computer vision and pattern recognition. 2022.
>
> [5] Jin, Peng, et al. "Expectation-maximization contrastive learning for compact video-and-language representations." Advances in neural information processing systems 35 (2022): 30291-30306.

---

### Note · Authors · 2025-08-11

**Final Remarks**

We sincerely thank all reviewers for their valuable suggestions and for recognizing the motivation and novelty of our work, which underscores the significance and value of both the problem we address and our proposed solution. In our rebuttal, we further supplemented technical details, theoretical derivations, error analysis, and image-domain generalization experiments to demonstrate the method’s effectiveness and generalization.

1. Regarding Reviewer ztES’s concern about real-world deployment and scalability to large-scale scenarios, we acknowledge the importance of this point. However, we note that in text–video retrieval (T2V) research, many highly influential methods (as listed in our reply) similarly break the dual-tower paradigm by introducing pair-wise cross-modal interactions, since the main challenge in this field lies in addressing information asymmetry and improving semantic alignment between modalities. Large-scale ANN-style retrieval is not yet the primary focus of the community. In this light, we feel this concern may not fully reflect the established research emphasis, and thus may not be entirely fair to our method. Nevertheless, we proposed a practical solution for applying GARE to ANN-based retrieval and provided error-bound analysis to verify its feasibility. We would be very grateful if this context could be considered in the final decision.

2. For Reviewer FbVo, we provided detailed efficiency analysis, adaptivity evaluation, and image-domain generalization analysis, addressing concerns on efficiency and generalization. While we did not introduce an adaptivity variant outperforming our best setting, we analyzed why a dual-sided adaptive design can reduce performance. Importantly, even this variant still outperforms the baseline, indicating meaningful adaptivity. We appreciate that Reviewer FbVo maintained a positive score.

3. For Reviewer 3THR, we responded point-by-point to questions on error analysis, efficiency, hard-negative handling, and the interaction between our three regularization losses. However, as the reviewer only provided an acknowledgement without further comments, we are not certain whether our responses fully addressed all the raised concerns.

4. For Reviewer SveN, we addressed the technical detail clarifications, modality gap motivation, and issues related to hard negatives and inference. We are glad these responses resolved the concerns, and we appreciate that Reviewer SveN raised their score.

---

### Decision · Program_Chairs · 2025-09-17

**Decision:**

Accept (poster)

**Comment:**

Considering that most of the reviewers' concerns have been addressed in the post-rebuttal process, the paper received three BA and one BR. The paper offers a novel mechanism and solution with a solid theoretical foundation, and features rigorous experimental design alongside extensive and qualitative analysis. It is recommended to accept this paper.